# Scalable Meta-Learning via Mixed-Mode Differentiation

**Iurii Kemaev** [1]  **Dan A. Calian** [1]  **Luisa M. Zintgraf** [1]  **Gregory Farquhar** [1]  **Hado van Hasselt** [1]

## Abstract

Gradient-based bilevel optimisation is a powerful technique with applications in hyperparameter optimisation, task adaptation, algorithm discovery, meta-learning more broadly, and beyond. It often requires differentiating through the gradient-based optimisation process itself, leading to "gradient-of-a-gradient" calculations with computationally expensive second-order and mixed derivatives. While modern automatic differentiation libraries provide a convenient way to write programs for calculating these derivatives, they oftentimes cannot fully exploit the specific structure of these problems out-of-the-box, leading to suboptimal performance. In this paper, we analyse such cases and propose **Mixed-Flow Meta-Gradients**, or *MixFlow-MG* – a practical algorithm that uses mixed-mode differentiation to construct more efficient and scalable computational graphs yielding over 10x memory and up to 25% wall-clock time improvements over standard implementations in modern meta-learning setups.

## 1. Introduction

Bilevel optimisation (BLO) is a commonly used tool to solve problems in meta-learning and deep learning (Liu et al., 2021; Zhang et al., 2024). In this problem setting, an *inner-loop* optimisation of parameters $\theta$ incrementally searches for optimal values $\theta^*$, in a process that depends on (fixed) meta-parameters $\eta$. In an *outer-loop* meta-optimisation, we search for optimal meta-parameters $\eta^*$. For instance, $\eta$ may include hyperparameters of the inner update (Bengio, 2000) or even their per-weight versions (Sutton, 1992).

This framework offers a powerful approach to automating the design and optimisation of learning systems, leading to significant advancements in various machine learning domains. It has applications ranging from hyperparameter optimisation (Bengio, 2000; Franceschi et al., 2018), data weighting (Hu et al., 2023b; Calian et al., 2025), and fast task adaptation (Finn et al., 2017), to neural architecture search (Liu et al., 2018), adaptive reinforcement learning (Xu et al., 2018; Zahavy et al., 2020), algorithm discovery (Oh et al., 2020), and more.

In gradient-based bilevel optimization, the meta-parameter update requires backpropagating through the inner loop, leading to second-order derivatives (gradients of gradients) – a notoriously computationally expensive process both in terms of memory and FLOPs. Updating outer parameters every $T$ inner steps (truncated backpropagation through time, Truncated-BPTT; Werbos, 1990) still results in computational cost scaling linearly with $T$. Consequently, we are often restricted to small inner and outer models $\theta$ and $\eta$ and short horizons $T$, limiting the exploration of the full potential of BLO. While Truncated-BPTT can be effective for smaller meta-models $\eta$ (Xu et al., 2018; Shaban et al., 2019), its applicability to large neural networks with billions of parameters (Gemini et al., 2023; OpenAI et al., 2023) remains an open question. Moreover, given the demonstrated impact of scale on model performance (Kaplan et al., 2020; Hoffmann et al., 2022b), the trend of scaling inner models $\theta$ is likely to continue. This necessitates more efficient BLO algorithms to support modern and future generations of models and to explore larger backpropagation horizons $T$ whilst keeping the cost of experiments affordable.

In this paper, we first analyse standard implementations for Truncated-BPTT-based bilevel gradients in modern frameworks for automated differentiation and highlight their inherent inefficiencies. We then propose **Mixed-Flow Meta-Gradients**, or *MixFlow-MG* – a simple reparameterization of the inner-loop learning dynamics that exposes the underlying symmetry of the problem and uses mixed-mode automatic differentiation to seamlessly exploit it. Finally, we use modern hardware and libraries for tensor programming to demonstrate that the proposed algorithmic technique, whilst requiring only minor code modifications, yields significant performance improvements in common meta-learning scenarios. In a representative setting, *MixFlow-MG* **demonstrates reductions up to** $95\%$ **in the active memory consumption and** $25\%$ **reduction in wall-clock time**, thus *allowing to scale bilevel gradient setups by more than an order of magnitude in a compute-efficient way.*

[1]Google DeepMind. Correspondence to: Iurii Kemaev <iukemaev@google.com>.

*Proceedings of the $42^{nd}$ International Conference on Machine Learning*, Vancouver, Canada. PMLR 267, 2025. Copyright 2025 by the author(s).

While numerous approximations for the (Truncated-)BPTT-based numerical procedure, such as implicit (Rajeswaran et al., 2019; Lorraine et al., 2020; Blondel et al., 2022; Choe et al., 2023) and forward-mode gradients (Silver et al., 2021; Shen et al., 2024) have been proposed recently, we focus on calculating *exact* gradients to isolate and address the core computational bottlenecks. The presented ideas can be seamlessly incorporated into approximate methods as well.

## 2. Background

### 2.1. Bilevel Optimisation

In the general form, BLO can be posed as the following constrained optimisation problem:

$$\min_{\eta} V(\eta), \text{ where } V(\eta) = \mathbb{E}_{y \sim Y} V(\theta^*(\eta), y) \quad (1)$$

$$s.t. \ \theta^*(\eta) = \arg\min_{\theta} \mathbb{E}_{x \sim X} L(\theta, \eta, x) \quad (2)$$

where $\eta$ are the outer meta-parameters, $\theta$ are the inner model parameters, $V$ and $L$ are validation and train losses calculated on the data points $y \sim Y$ and $x \sim X$, respectively. Note that standard network training regimes are a special case, where the validation loss in Equation (1) is minimised by tuning the meta-parameters $\eta$ by hand.

Typically $\theta^*(\eta)$ in Equation (1) is approximated with $T$ steps of gradient-based methods (Maclaurin et al., 2015):

$$\min_{\eta} V(\eta) := \mathbb{E}_{y \sim Y} V(\theta_T(\eta), y),$$
$$(\theta_{i+1}, \upsilon_{i+1}) = \Phi(\theta_i, \upsilon_i, \eta, x_i) \quad i = 0 \ldots T - 1 \quad (3)$$

where $\upsilon_i$ is an arbitrary state at step $i$, such as an optimiser's momentum, and $\Phi(\theta_i, \upsilon_i, \eta, x_i)$ is an update that involves calculating the gradient $\partial L(\theta_i, \eta, x_i)/\partial \theta_i$ and is differentiable by $\eta$. This ensures that meta-parameters $\eta$, in their turn, can also be optimised with gradient methods, giving rise to quantities involving second-order derivatives of the loss function $L(\theta, \eta, x)$. In particular, such schemes require computing left- or right-hand side products of the second-order derivatives with arbitrary vectors.

### 2.2. Primer on Automatic Differentiation

A convenient way to compute the quantities involving second-order derivatives in Equation (3) is provided by modern automatic differentiation libraries such as JAX (Bradbury et al., 2018) or PyTorch (Paszke et al., 2017). This section explains fundamental concepts upon which these libraries are built, which is important for understanding how to design efficient algorithms for solving equations (3).

Let us consider arbitrary $f(x) : \mathbb{R}^n \to \mathbb{R}^m$ with the corresponding Jacobian $J = \partial f/\partial x \in \mathbb{R}^{m \times n}$. Autodiff provides two types of differentiation for such functions: forward

and reverse. Forward mode calculates Jacobian-by-vector product (JVP) $Jv$ with arbitrary vector $v$ at a computational cost proportional to a single forward pass (Baur & Strassen, 1983). By carrying out JVPs with $n$ input's basis vectors, the full Jacobian $J$ can be recovered column-by-column, hence requiring $O(n)$ forward passes in total. Reverse mode, on the other hand, computes vector-by-Jacobian product (VJP) $\nu J$ and recovers the Jacobian one row at a time, in total requiring $O(m)$ forward passes for computing the full Jacobian; however, by design, it operates in two passes – forward and backward – and requires storing all intermediate activations during the forward pass to use them in the backward pass, resulting in significantly higher memory requirements.

For neural networks, typical objects for differentiation are loss functions $L : \mathbb{R}^n \to \mathbb{R}$ that output scalars. This is why reverse mode is the default choice, since it recovers the whole Jacobian $J_L$ in $O(1)$ forward passes.

Certain classes of differential programs, such as those implementing second-order optimisation in Section 2.1, require computing products with second-order derivatives of the corresponding loss functions. One example is Hessian-by-vector products (HVPs) $\partial^2 L/\partial x^2 v$. HVPs can be cheaply evaluated using repeated VJP and/or JVP products (Pearlmutter, 1994), and there are three computationally tractable ways available in practice:

$$\frac{\partial^2 L}{\partial x^2} v = e \frac{\partial}{\partial x} \left( \frac{\partial L}{\partial x} v \right) = \underbrace{VJP(e, JVP(L, v))}_{\textbf{reverse-over-forward}}$$

$$= \frac{\partial}{\partial x} \left( e \frac{\partial L}{\partial x} \right) v = \underbrace{JVP(VJP(e, L), v)}_{\textbf{forward-over-reverse}}$$

$$= v \frac{\partial}{\partial x} \left( e \frac{\partial L}{\partial x} \right) = \underbrace{VJP(v, VJP(e, L))}_{\textbf{reverse-over-reverse}},$$

where $v \in \mathbb{R}^n$ is an arbitrary vector and $e = (1)_{1 \times 1}$ is a unit vector. Note that *forward-over-forward* mode was purposefully omitted due to its prohibitive computational cost of $O(n)$ forward passes.

One crucial observation is that *forward-over-reverse* mode avoids storing activations from the inner backward pass, often making it the most memory efficient choice in practice. In addition, it has lower I/O overhead (no need to read/write activations), potentially leading to reductions in wall-clock time. This advantage becomes even more apparent when calculating $\partial L/\partial x$ relies on the gradient checkpointing technique (Griewank & Walther, 2000), as it is effectively a no-op for forward-mode differentiation. This property forms the core of a highly efficient algorithm for gradient-based BLO which will be described further in the paper.

## 2.3. Automatic Differentiation in BLO

Several works (e.g., Franceschi et al., 2017) explore the trade-offs between forward- and reverse-mode differentiation for meta-parameters $\eta$ in gradient-based BLO in Equation (3). In this context, the computational cost of reverse-mode differentiation for the validation loss $J_V$ with respect to $\eta$ is comparable to the computation of the inner- and outer losses themselves, i.e. $O(|\eta|) + O(T|\theta|)$, but requires storing all intermediate activations in memory that are not necessary in forward-mode differentiation. However, this memory efficiency comes at the price of the increased computational cost, which becomes $O(|\eta| \times |\theta|)$. The overall consensus is that when the number of meta-parameters is small, one should consider forward-mode differentiation to avoid incurring extra memory costs. However, in many modern applications (Finn et al., 2017; Wichrowska et al., 2017; Oh et al., 2020), $|\eta|$ can be much larger, often even comparable in size to the number of model parameters $|\theta|$ (e.g., $\eta$ are themselves the parameters of a neural network, as in Finn et al. (2017)), thus making forward-mode differentiation prohibitively expensive. In such cases, a straightforward implementation uses reverse mode at both levels of the corresponding program, as shown in Algorithm 1. All of the open-source repositories for gradient-based meta-optimisation that we verified use this implementation (e.g., Arnold et al., 2020; Metz et al., 2022). We therefore refer to Algorithm 1 as a *standard* or *default* autodiff-based implementation further in the text.

The aforementioned autodiff libraries (Paszke et al., 2017; Bradbury et al., 2018) compile the computational graph that is defined by a user's program (in just-in-time or dynamic regime) before executing it. Compilation allows for leveraging advanced optimisations and memory allocation algorithms, which can make all three differentiation modes theoretically equivalent in terms of compute- and memory efficiency in many use cases (Dagréou et al., 2024; Blondel & Roulet, 2024).

Nevertheless, their corresponding hidden constants can substantially differ in practice. The practical differences stem from various factors, including: the underlying model's structure, inputs' size, autodiff framework and hardware backend, compiler's configuration and flags, custom optimisations, etc. Moreover, these frameworks only have limited contextual information about a given problem's nature, hence often failing to capture and exploit inherent symmetries and structural bottlenecks of the problem at hand, which can lead to suboptimal low-level programs in practice, as we will demonstrate further in the paper.

In the following section we identify an algorithmic improvement based on the fact that a standard computational graph for bilevel gradients includes symmetric matrices, such as Hessians, which are never fully instantiated or explicitly

defined in the code. By exploiting their hidden symmetry in a non-intrusive way, we achieve substantial memory savings with minimal code changes in the user programs. Our benchmarks show that modern compilers are not able to make such improvements on their own.

# 3. *MixFlow-MG*: Mixed-mode Differentiation for Bilevel Gradients

We are now going to decompose equations (3) in order to expose the Hessian matrix. This will allow us to employ a more memory-efficient algorithm for calculating the outer gradients.

Firstly, we propose reparameterising equations (3) to have gradients $\nabla L_i = \partial L(\theta_i, \eta, x_i)/\partial \theta_i$ as a separate argument in the combined update function $\Upsilon$:

$$(\theta_{i+1}, v_{i+1}) = \Phi(\theta_i, v_i, \eta, x_i) = \Upsilon(\nabla L_i, \theta_i, v_i, \eta, x_i). \quad (4)$$

After applying the chain rule to the gradient of the validation loss with respect to $\eta$ and using the fact that the validation loss $V$ in (3) does not depend on the last-step state $v_T$, we obtain (in vector notation)

$$\frac{\mathrm{d}V}{\mathrm{d}\eta} = \frac{\partial V}{\partial \theta_T}\frac{\mathrm{d}\theta_T}{\mathrm{d}\eta} + \frac{\partial V}{\partial v_T}\frac{\mathrm{d}v_T}{\mathrm{d}\eta} = \begin{pmatrix} \frac{\partial V}{\partial \theta_T} & 0 \end{pmatrix}\frac{\mathrm{d}\Upsilon_T}{\mathrm{d}\eta}. \quad (5)$$

Then, after unrolling one step for $\Upsilon_{i+1}$ (Appendix A.2), we get the following recurrent relation for the total derivatives:

$$\frac{\mathrm{d}\Upsilon_{i+1}}{\mathrm{d}\eta} = \left( \frac{\partial \Upsilon_{i+1}}{\partial \theta_i} + \frac{\partial \Upsilon_{i+1}}{\partial \nabla L_i}\frac{\partial^2 L}{\partial \theta_i^2} \quad \frac{\partial \Upsilon_{i+1}}{\partial v_i} \right)\frac{\mathrm{d}\Upsilon_i}{\mathrm{d}\eta} + \\ + \frac{\partial \Upsilon_{i+1}}{\partial \nabla L_i}\frac{\partial^2 L}{\partial \eta \partial \theta_i} + \frac{\partial \Upsilon_{i+1}}{\partial \eta}. \quad (6)$$

Equation (6) allows to unroll the loop "backwards", from $i = T - 1$ to $0$. According to Equation (5), for calculating $\mathrm{d}V/\mathrm{d}\eta$ it needs to be multiplied by the vector $\begin{pmatrix} \partial V/\partial \theta_T & 0 \end{pmatrix}$ from the left, hence it only requires one VJP. However, it can be noticed that Equation (6) contains explicit vector-by-hessian and vector-by-mixed-derivatives-matrix products; the default autodiff implementation will perform them in reverse-over-reverse mode, which can be suboptimal in practice.

To circumvent this, we transform the relation using classical results. Assuming that the function approximator and loss function have continuous second-order derivatives, which is typically the case for neural networks, the following identities hold (c.f. *Schwarz's theorem*):

$$\frac{\partial^2 L}{\partial \theta_i^2}^T = \frac{\partial^2 L}{\partial \theta_i^2}, \quad \frac{\partial^2 L}{\partial \eta \partial \theta_i}^T = \frac{\partial^2 L}{\partial \theta_i \partial \eta}.$$

| **Algorithm 1** Standard Truncated-BPTT (Equation (3)) | **Algorithm 2** Mixed-mode Truncated-BPTT (Equation (4)) |
|---|---|
| **Input:** $\eta, \theta_0, \upsilon_0$, inputs $\{x_i\}_{t=1}^T$, $val\_x$ | **Input:** $\eta, \theta_0, \upsilon_0$, inputs $\{x_i\}_{t=1}^T$, $val\_x$ |
| **Output:** $\partial V/\partial \eta$ | **Output:** $\partial V/\partial \eta$ |
| 1: | 1: |
| 2: **function** $\Phi(\theta, \upsilon, \eta, x_i)$ | 2: **function** $\Upsilon(\nabla L, \theta, \upsilon, \eta, x_i)$ |
| 3:    $\nabla L \leftarrow grad(L)(\theta, x_i)$ | 3:    *empty line* |
| 4:    ... arbitrary operations on $\theta, \upsilon, \nabla L$ | 4:    ... arbitrary operations on $\theta, \upsilon, \partial\theta$ |
| 5:    $(\theta, \upsilon) \leftarrow optimizer(\theta, \upsilon, \nabla L)$ | 5:    $(\theta, \upsilon) \leftarrow optimizer(\theta, \upsilon, \partial\theta)$ |
| 6:    ... arbitrary operations on $\theta, \upsilon$ | 6:    ... arbitrary operations on $\theta, \upsilon$ |
| 7:    **return** $\theta, \upsilon$ | 7:    **return** $\theta, \upsilon$ |
| 8: | 8: |
| 9: **function** VALLOSS$(\eta, \theta_0, \upsilon_0, \{x_i\}_{t=1}^T, val\_x)$ | 9: **function** VALLOSS$(\eta, \theta_0, \upsilon_0, \{x_i\}_{t=1}^T, val\_x)$ |
| 10:   $(\theta, \upsilon) \leftarrow (\theta_0, \upsilon_0)$ | 10:   $(\theta, \upsilon) \leftarrow (\theta_0, \upsilon_0)$ |
| 11:   **for** $i \leftarrow 1$ to $T$ **do** | 11:   **for** $i \leftarrow 1$ to $T$ **do** |
| 12:     *empty line* | 12:     $\nabla L \leftarrow$ **fwdrev_grad(L)**$(\theta, x_i)$ |
| 13:     $(\theta, \upsilon) \leftarrow \Phi(\theta, \upsilon, \eta, x_i)$ | 13:     $(\theta, \upsilon) \leftarrow \Upsilon(\nabla L, \theta, \upsilon, \eta, x_i)$ |
| 14:   **return** $V(\theta, val\_x)$ | 14:   **return** $V(\theta, val\_x)$ |
| 15: | 15: |
| 16: $\partial V \leftarrow$ grad(ValLoss)$(\eta, \theta_0, \upsilon_0, \{x_i\}_{t=0}^{T-1}, val\_x)$ | 16: $\partial V \leftarrow$ grad(ValLoss)$(\eta, \theta_0, \upsilon_0, \{x_i\}_{t=0}^{T-1}, val\_x)$ |
| 17: **return** $\partial V$ | 17: **return** $\partial V$ |

Combining them with Equation (6) we can rewrite the vector-by-hessian (VHP) and vector-by-mixed-derivatives-matrix (VMP) products into their transposed versions, i.e. hessian-by-vector (HVP) and mixed-derivatives-matrix-by-vector (MVP) products:

$$\underbrace{v\frac{\partial \Upsilon_{i+1}}{\partial \nabla L_i}\frac{\partial^2 L}{\partial \theta_i^2}}_{\textbf{inefficient VHP}} = \left(\underbrace{\frac{\partial^2 L}{\partial \theta_i^2}\overbrace{\left(v\frac{\partial \Upsilon_{i+1}}{\partial \nabla L_i}\right)^T}^{\textit{normal VJP}}}_{\textbf{efficient HVP}}\right)^T \qquad (7)$$

$$\underbrace{v\frac{\partial \Upsilon_{i+1}}{\partial \nabla L_i}\frac{\partial^2 L}{\partial \eta \partial \theta_i}}_{\textbf{inefficient VMP}} = \left(\underbrace{\frac{\partial^2 L}{\partial \theta_i \partial \eta}\overbrace{\left(v\frac{\partial \Upsilon_{i+1}}{\partial \nabla L_i}\right)^T}^{\textit{normal VJP}}}_{\textbf{efficient MVP}}\right)^T . \qquad (8)$$

**Proposition 3.1.** *Reparamererisation* (4) *and identities* (7), (8) *allow for replacing the default reverse-over-reverse differentiation for recurrent relation* (6) *with more efficient forward-over-reverse or reverse-over-forward alternatives.*

Since mixed-mode differentiation constitutes the core algorithmic improvement in this technique, we call it **Mixed-Flow Meta-Gradients** or *MixFlow-MG*.

While advanced autodiff compilers and memory allocation algorithms can make all three differentiation modes equivalent in terms of compute- and memory efficiency in most

of cases (Dagréou et al., 2024; Blondel & Roulet, 2024), their practical performance can vary remarkably, which we demonstrate in Section 5. In general case, it is recommended trying all three options for choosing the best one for a setup at hand; the proposed reparameterisation (4) makes this probing straightforward. In Section 5 we demonstrate how *MixFlow-MG* leverages forward-over-reverse differentiation for significant performance gains in practice.

### 3.1. Implementation in JAX

JAX (Bradbury et al., 2018) is a powerful library for differential tensor programming. It relies on the functional paradigm, where stateless functions can be transformed and returned by other functions; one of its key transformations is $grad(f)$ which accepts a scalar-valued function $f(x): \mathbb{R}^n \to \mathbb{R}$ and returns a new function $g(x): \mathbb{R}^n \to \mathbb{R}^n$ that computes the gradient of $f$ with respect to $x$, i.e. $g(x) \coloneqq \partial f/\partial(x)$.

The default autodiff-based implementation uses this convenient notation for computing second-order derivatives in the original training loop (3), as shown in Algorithm 1. This however can be highly suboptimal as it fails to exploit the problem's inherent symmetries, as discussed in Section 2.3. Our proposed reparameterisation (4), outlined in Algorithm 2, allows to use mixed-mode differentiation via custom *fwdrev_grad* transformation, which defines a VJP rule for calculating HVPs in forward-over-reverse mode. This requires changing only a few lines of code; our implementation can be found in Appendix A.4.

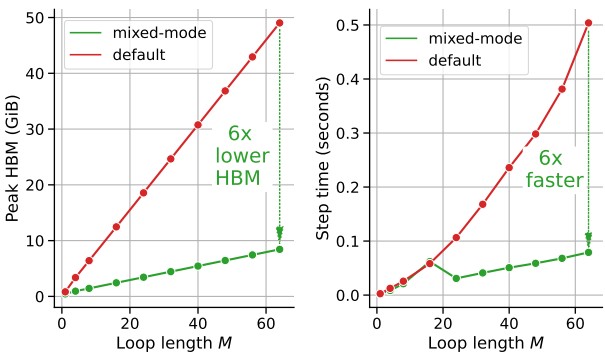

*Figure 1.* Peak HBM and step time across the number of (per inner step) transformations $M$ in Equation (9) (GPU).

### 3.2. Motivating Example

To illustrate the effects of *MixFlow-MG*, we consider the following simple BLO problem (Equation (3)), similar to Finn et al. (2017): $\eta$ defines the initialisation point $\theta_0 = \eta$ for the inner optimisation; the inner loss is a standard L2 loss which is independent of $\eta$; the update dynamics $\Phi(\theta_i, \upsilon_i, \eta, x_i)$ is a standard stateless ($\upsilon = \varnothing$) gradient step.

The inner model $y_M$ is the following $M$-step recursive map:

$$y_i(\theta, x) = i \cdot \left(2 + sin(y_{i-1})\right)^{cos(y_{i-1})}, \qquad (9)$$

where $y_0(\theta, x) = \theta x$, $x \in \mathbb{R}^{B \times D}$, $\theta \in \mathbb{R}^{D \times D}$. We used $B = 1024$ and $D = 4096$ in our experiments and kept the number of inner updates $T = 2$ for simplicity. The computational graph gets longer as the number of (per inner step) transformations $M$ increases, meaning we can study the effects on memory and runtime by adjusting $M$. For the sake of demonstration, we minimised the effects of compiler's optimisation by disabling all loop fusions.

Figure 1 demonstrates how the metrics change across the number of per-step transformations $M$. The HBM and step-time scales much better when using mixed-mode differentiation, with memory and wall-clock reductions up to $85\%$ as $M$ increases. The corresponding code and XLA-generated computational graphs are given in Appendices A.6 and A.7.

## 4. Scaling to Large Models

This section investigates device memory patterns and memory optimisation techniques in gradient-based bilevel optimisation for the case when the underlying models get larger.

The standard implementation of Truncated-BPTT for BLO (Algorithm 1) loops over $T$ inner updates $\Phi$ to obtain $\theta_T$ for calculating the outer (validation) loss. If done naively, this algorithm requires storing intermediate activations $A_t$ and outputs $\theta_t$, $\upsilon_t$ for each of $t = 1..T$ inner steps, hence the peak memory consumption for one meta update scales

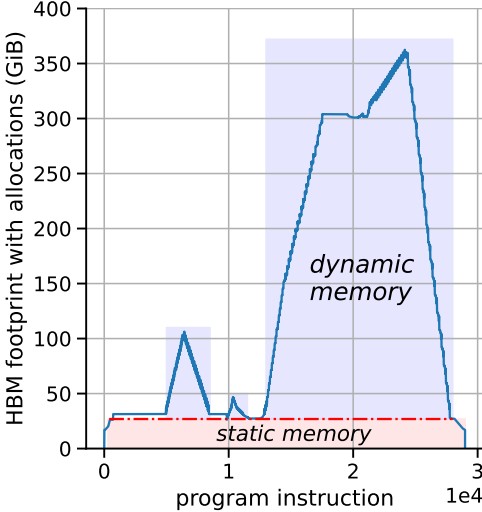

*Figure 2.* Device memory footprint for an outer update when using autodiff for one step of bilevel optimisation. The memory can be divided into static (checkpoints, inputs, parameters, states) and dynamic (activations for backpropagation). The dynamic memory can be reduced by exploiting the problem structure (see Figure 3). X-axis: instruction number in a compiled computation graph. Y-axis: the amount of occupied device memory.

as $O(T \cdot (|A| + |\theta| + |\upsilon|))$. While it can be affordable for small setups, real-world models are too large to be adapted for meta-training this way due to high cost and scarcity of the high bandwidth on-device memory: typically, one inner step already uses all available on-accelerators memory.

Gradient checkpointing (Griewank & Walther, 2000) for inner steps is often used in practice (e.g. in Arnold et al. (2020); Metz et al. (2022)) to bring the memory footprint down to $O(|A| + T \cdot (|\theta| + |\upsilon|))$, since only activations for the current step are kept in memory at any moment of time during meta-backpropagation, and $T \cdot (|\theta| + |\upsilon|)$ parameters are getting checkpointed during the outer-loop unroll. Typically the size of activations and partial derivatives $|A|$ is substantially larger than the size of parameters and optimiser states $|\theta| + |\upsilon|$ due to the dependency of the former on both the latter and inputs' sizes. This makes gradient checkpointing instrumental for scaling, and following this common practice, we keep it enabled in all our benchmarks.

One important distinction to make is that checkpoints, inputs, parameters, and states require $O(T \cdot (|\theta| + |\upsilon|))$ bytes that get allocated at the beginning of a program for the whole execution time and written to only once. For this reason, we refer to this type of memory as ***static***. On the contrary, $O(|A|)$ bytes are allocated during outer-level backpropagation and re-purposed for new activations at every inner step, hence we refer to it as ***dynamic*** memory. Typical memory footprint for a single outer step can be found in Figure 2.

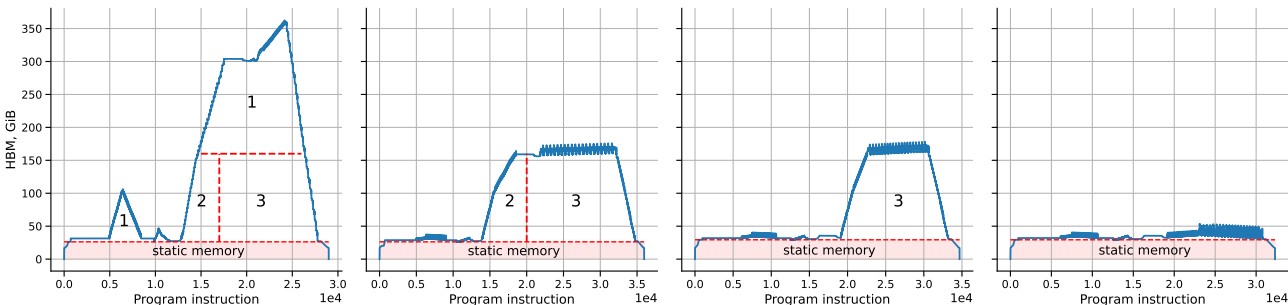

*Figure 3.* HBM footprints for each stage of optimisation for 489M chinchilla transformer on GPU: 1 – **block rematerialization**, 2 – **saving inner gradients**, 3 – **mixed-mode differentiation**. Each optimisation is responsible for reducing a specific chunk of HBM.

In addition to enabling gradient checkpointing for inner loop unrolling, we found the following two optimisations important for amplifying the gains of the proposed algorithm:

1. ***Block rematerialisation***: for neural networks with block-residual connections $x_{i+1} = f_i(x_i) + x_i$, such as residual networks (He et al., 2016) and transformers (Vaswani, 2017), gradient checkpointing can be applied to each of the blocks $f_i$ to substantially reduce memory footprint at the theoretical cost of one forward pass; this is a known optimisation, hence we keep it enabled for both baseline and the proposed method to avoid running out of memory even for smallest networks.

2. ***Saving inner gradients***: $\partial L/\partial \theta$ can be saved (in addition to per-inner step inputs and parameters $\theta$) as part of inner-loop gradient checkpointing to avoid incurring one extra backward pass during the outer-level gradient propagation; we have not found this optimisation in previous works and existing libraries, hence it can be considered as an additional contribution of this paper; we enable it only for *MixFlow-MG*.

Both these optimisations plus mixed-mode differentiation, as introduced in Section 3, complement each other. Figure 3 include the ablation study for 489M Chinchilla model (on MAML; our full benchmark setup is described in the next section). In particular, block rematerialisation under forward-over-reverse differentiation does not require storing intermediate per-block checkpoints thanks to the forward mode at the outer level. This allows to almost completely remove block #3 in Figure 3 thus drastically reducing peak memory usage. Note that some portion of extra memory is still claimed for calculating activations and JVPs on-the-fly, this is why forward mode differentiation typically requires 3 times more memory than the basic forward pass.

We also observed that, while saving inner gradients is crucial both for memory and step-time reductions on GPUs, it was only important for the latter on TPUs, which shows the dif-

ference in the compilers' efficiency for these two backends. More details on this can be found in Appendix A.8.

## 5. Benchmarking Language Modelling Tasks

The primary goal of this section is to demonstrate the benefits of *MixFlow-MG* on a representative set of BLO setups. Without limiting generalisation, we chose the language modelling domain for the inner-level optimisation, where the standard loss is the next-token-prediction loss $NTP(\theta, x)$. We use the *Chinchilla* family of language models (Hoffmann et al., 2022a) with RoPE (Su et al., 2024) and the Adam optimiser (Kingma, 2014). When a meta model is present, we use the same architecture as for the inner model.

Firstly, we explain the rationale behind choosing the performance metrics. Then, we select a diverse class of real-world problems to demonstrate possible gains in practice. Further, we investigate different properties of *MixFlow-MG* using various model structures and data regimes. Finally, combining all findings, we provide practical recommendations on efficiently scaling bilevel gradient setups by orders of magnitude beyond any existing frameworks.

Benchmarking was performed in JAX (Bradbury et al., 2018) on TPUv5p and H100 using the OpenXLA backend and libraries from DeepMind et al. (2020). While we observe consistent behaviour across setups and tasks, results may vary depending on library versions, hardware, compiler flags, and other factors beyond the scope of this work.

We listed the minimal changes required for implementing *MixFlow-MG* in Section 3.1 and included the relevant Python code for JAX and PyTorch in Appendix A.4.

### 5.1. Metrics

*MixFlow-MG* operates on a per-inner-step basis, i.e. it addresses dynamic memory. In our metrics we focus solely on dynamic memory and defer to Appendix A.3 for practical recommendations on how to reduce static memory.

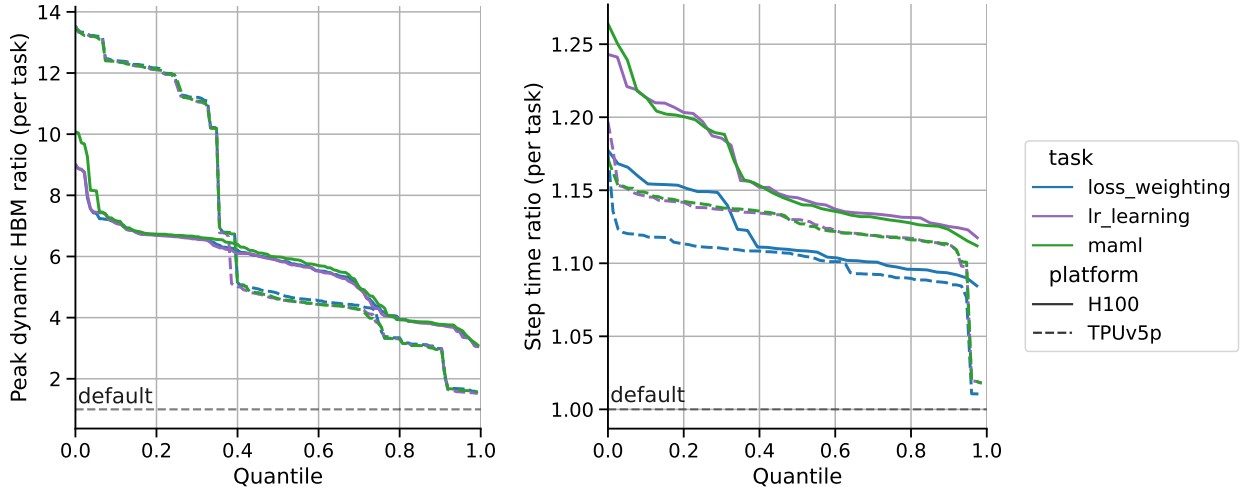

Figure 4. Joint sweep over tasks, models, and hyperparameters from Table 1: peak dynamic HBM and step time ratio between default and mixed-mode differentiation, sorted by value in descending order (higher is better, and values > 1 mean that *MixFlow-MG* improves over the default autodiff implementation). All variations win both memory- and compute-wise, with highly correlated gains between tasks.

Table 1. Sweep over tasks: hyperparameters and values.

| Parameter | Values |
|---|---|
| Task | {learning_lr, maml, loss_weighting} |
| Model size ($\times 10^6$) | {57, 106, 163, 217, 306} |
| # of inner updates ($T$) | {2, 4, 8} |
| Batch size | {2, 4, 8} |
| Sequence length | {2048, 4096, 8192} |

We measure peak dynamic High Bandwidth Memory (HBM) (device memory) and wall-clock step time. Where more appropriate, we report two performance metrics which are defined as a ratio of the corresponding measurements between the default implementation and the proposed changes, i.e. higher values indicate stronger gains over the baselines.

**Peak dynamic HBM ratio** is the ratio between the peak usages of **dynamic** HBM (Section 4)

$$\frac{HBM_{\text{default}} - HBM_{\text{default}}^{\text{static}}}{HBM_{\text{MixFlow-MG}} - HBM_{\text{MixFlow-MG}}^{\text{static}}}. \quad (10)$$

**Step Time ratio** is the ratio between wall-clock time per meta step

$$T_{\text{default}} / T_{\text{MixFlow-MG}}. \quad (11)$$

### 5.2. Sweeping over Bilevel Optimisation Tasks

To recap, a typical setup for the gradient-based BLO Equation (3) is comprised of an inner loop that updates model parameters $\theta$ for $T$ steps, and an outer loop that updates $\eta$ by backpropagating $\partial V / \partial \eta$ through the inner loop steps by unrolling it backwards; the particular dependence of the inner-loop optimisation on $\eta$ defines the problem setup.

We consider the following three common BLO setups:

- **Hyperparameter Learning**: similar to Bengio (2000) and Maclaurin et al. (2015), $\eta$ are the per-parameter learning rates for the inner optimiser, so that

$$\theta_{i+1} = g(\eta, \partial NTP(\theta_i, x_i)/\partial \theta_i, \theta_i, \upsilon_i),$$

  with $g$ a function that includes optimiser's transformations for converting gradients into parameter updates.

- **Model-Agnostic Meta-Learning** (MAML, Finn et al. (2017)): $\eta$ defines the initialisation point $w_0 = \eta$ for the inner optimisation and the inner loss is otherwise independent of $\eta$:

$$L(\theta_i, \eta, x_i) = NTP(\theta_i, x_i).$$

- **Meta-learning Adaptive Loss Weighting**: inspired by Hu et al. (2023a), this setup uses $\eta$ to calculate per-data point loss weighting factors:

$$L(\theta_i, \eta, x_i) = \alpha(\eta, x_i) \cdot NTP(\theta_i, x_i).$$

We sweep over the hyperparameters in Table 1, totalling in 135 distinct configurations per task and sort all results by gains, in descending order. Figure 4 shows memory gains and step-time wins for the runs that fit in available device memory (80 GiB for GPU and 96 GiB for TPU).

*MixFlow-MG* delivers substantial improvements across the board. We observe that memory footprint and step time are reduced for *all* hyperparameter combinations. Remarkably, memory usage is decreased by approximately 75% (*nearly*

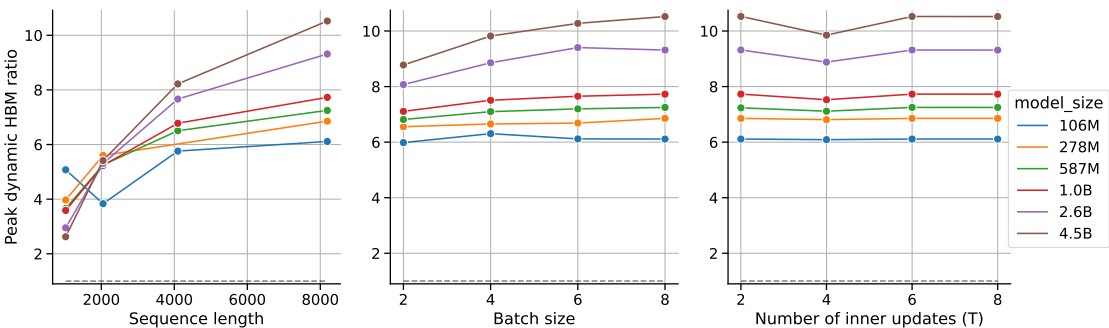

*Figure 5.* Sweep over data regimes for Chinchilla models (GPU): peak dynamic HBM ratio between default and mixed-mode diff-n.

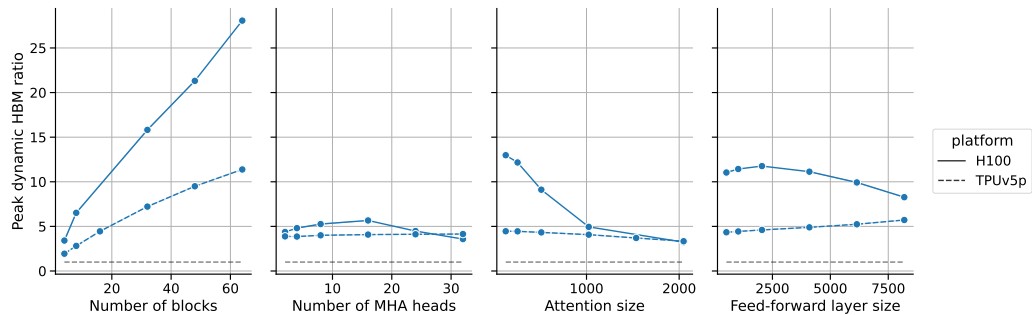

*Figure 6.* Sweep over transformer components: peak dynamic HBM ratio between default and mixed-mode differentiation.

*4x less memory*) for 80% of configurations, with peak reductions exceeding 90% (*over 10x less memory*) on both GPUs and TPUs. Previously, memory constraints severely limited the scale of bilevel optimization. These results open the door to training models of much larger size and complexity. Wall-clock time wins reach 25% for GPU and 20% for TPU, with a median improvement of 12% for both.

Wall-clock gains are almost uniform across configurations, while memory gains vary significantly. We investigate this in the following, and disentangle factors contributing to the memory behaviour to showcase *MixFlow-MG*'s properties.

### 5.3. Model and Data Scaling

The dynamic memory requirements of transformer models using the default implementation scale as $O(BL(S+kS^2))$, where $L$ is the number of layers, $S$ the context length, $B$ the batch size, and $k$ a compiler-dependent constant. This scaling arises from the self-attention blocks and holds even with block rematerialisation enabled in the default implementation. However, as detailed in Section 4, our proposed mixed-mode differentiation with block rematerialisation offers a significant advantage: its memory usage is independent of the number of layers, scaling only as $O(B(S+\hat{k}S^2))$, where $\hat{k}$ represents the corresponding constant for mixed-mode gradients and is significantly smaller than $k$. This reduction stems from the forward-over-reverse mode, which requires

only a single memory buffer for activations, as opposed to number of blocks buffers for the default implementation.

This difference in memory scaling leads to a substantial reduction in peak dynamic HBM usage, quantified by the ratio:

$$\frac{BL(S+kS^2)}{B(S+\hat{k}S^2)} = \frac{L(1+kS)}{1+\hat{k}S}. \quad (12)$$

The factor $L$ in the enumerator ensures that *MixFlow-MG* is *an algorithmic improvement* for models with block-residual connections, such as residual networks and transformers.

To validate this theoretical estimate, we benchmark combinations of transformer models, context lengths, batch sizes, and number of updates. In the previous section we observed that *MixFlow-MG* shows highly correlated gains across all tasks, so we report metrics only for the MAML setup here.

Figure 5 shows the gains for different models, batch sizes $B$, context lengths $S$, and inner-loop lengths $T$ for GPUs, with similar dynamics observed for TPUs. These empirical results closely align with Equation (12): discounting minor compilation effects, the gains are constant across $B$ and $T$ and sub-linearly increase towards $kL/\hat{k}$ for larger $S$.

The impact of scaling different architectural components of a Chinchilla transformer is shown in Figure 6. The memory gains scale linearly with the number of layers $L$, confirming

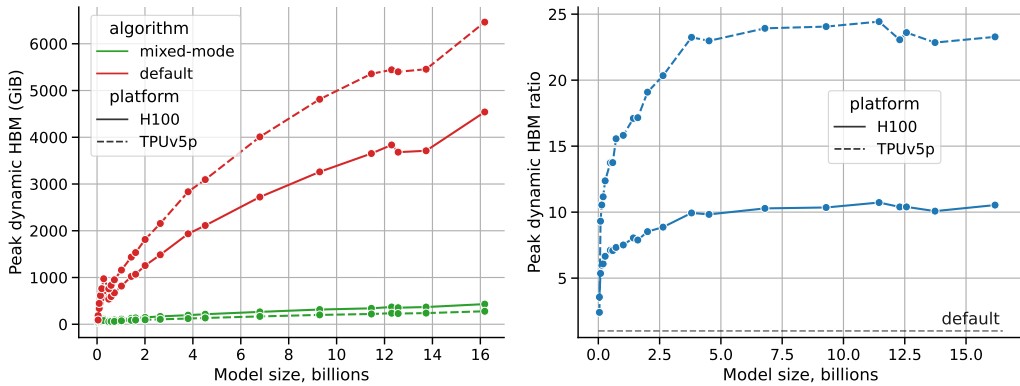

*Figure 7.* Chinchilla scaling ladder: peak dynamic HBM gains across transformers of various sizes.

our theoretical analysis. While the gains could be expected to be near-constant for the other structural parameters, the real numbers differ in practice, especially for small models on GPUs. This can be attributed to compilation effects: the smaller a computational graph, the more memory optimisations a low-level compiler can find in limited time, e.g. GPUs may be able to schedule the fixed-size thread warps more efficiently for small graphs.

In the real world transformers simultaneously scale across all components (Hoffmann et al., 2022a). Figure 7 shows the peak dynamic HBM gains across a reduced version of the original Chinchilla scaling ladder, with models ranging from 44M to 16B parameters. We observe that the gains get larger for bigger models, eventually converging to 23-25x (96%) dynamic memory reductions for TPUs and 10x (90%) for GPUs. We hypothesise that the convergence happens due to the underspecified compiler's behaviour given the fact that starting from 1B transformers, the corresponding default computational graphs outgrow any available memory by more than one order of magnitude, which can be too far from typical compilation targets.

## 6. Conclusion

In this paper, we examined the practical aspects of gradient-based methods for bilevel optimisations, identifying inefficiencies in default autodiff-based implementations. To address them, we proposed *MixFlow-MG* that uses mixed-mode differentiation for the most computationally demanding components in the corresponding programs. We achieved this by introducing a simple generic reparameterisation technique that can be effortlessly integrated into standard implementations. We conducted detailed analysis of the proposed algorithm and identified its scaling properties. Our empirical benchmarks demonstrated significant practical improvements, including up to 10x total memory reductions and 25% lower wall-clock time in modern

meta-learning setups. Importantly, as the domain models become larger and more sophisticated, the positive effect of using *MixFlow-MG* compounds, allowing to drastically reduce scaling costs. We believe that our work will help to facilitate research in gradient-based bilevel optimisation by opening up a larger class of models for experimenting whilst minimising the need for extra computational resources. We included a minimalistic implementation in JAX and PyTorch for *MixFlow-MG* in Appendix A.4 for reference and easy adoption.

## Acknowledgements

Authors would like to express their deep gratitude to Junhyuk Oh, Matteo Hessel, Dan Horgan, the JAX, XLA, and RL teams, and David Silver for fruitful discussions and support throughout the project. We also thank the reviewers for the valuable feedback that helped to improve the clarity of the paper.

## Impact Statement

This paper presents work whose goal is to advance the field of Machine Learning. There are many potential societal consequences of our work, none which we feel must be specifically highlighted here.

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

# A. Appendix

## A.1. Author Contributions

**Iurii Kemaev**: MixFlow-MG concept, project leadership, algorithm and benchmarks design, implementation, and analysis; **Dan A. Calian**, **Luisa M. Zintgraf**, **Gregory Farquhar**: algorithm analysis, benchmarks design, testing implementation; **Hado van Hasselt**: advising the project, algorithm refinement and analysis.

All authors contributed to paper writing.

## A.2. Derivations for *MixFlow-MG*

To expose second-order derivatives in the update equations Equation (3), we propose reparameterising them to have gradients $\nabla L_i = \partial L(\theta_i, \eta, x_i)/\partial \theta_i$ as a separate argument in the combined update function $\Upsilon$:

$$(\theta_{i+1}, \upsilon_{i+1}) = \Phi(\theta_i, \upsilon_i, \eta, x_i) = \Upsilon(\nabla L_i, \theta_i, \upsilon_i, \eta, x_i). \tag{13}$$

After applying the chain rule to the gradient of the validation loss with respect to $\eta$ and using the fact that the validation loss $V$ in (3) does not depend on the last-step state $\upsilon_T$, we obtain

$$\begin{aligned}\frac{\mathrm{d}V}{\mathrm{d}\eta} &= \frac{\partial V}{\partial \theta_T}\frac{\mathrm{d}\theta_T}{\mathrm{d}\eta} + \underbrace{\frac{\partial V}{\partial \upsilon_T}\frac{\mathrm{d}\upsilon_T}{\mathrm{d}\eta}}_{=\,0} = \begin{pmatrix}\frac{\partial V}{\partial \theta_T} & 0\end{pmatrix}\begin{pmatrix}\frac{\mathrm{d}\theta_T}{\mathrm{d}\eta}\\[4pt]\frac{\mathrm{d}\upsilon_T}{\mathrm{d}\eta}\end{pmatrix}.\\[4pt] &= \begin{pmatrix}\frac{\partial V}{\partial \theta_T} & 0\end{pmatrix}\frac{\mathrm{d}\Upsilon_T}{\mathrm{d}\eta}\end{aligned} \tag{14}$$

To calculate this total derivative, let us unroll one step for for $\Upsilon_{i+1}$:

$$\begin{aligned}\frac{\mathrm{d}\Upsilon_{i+1}}{\mathrm{d}\eta} &= \frac{\partial \Upsilon_{i+1}}{\partial \nabla L_i}\frac{\mathrm{d}\nabla L_i}{\mathrm{d}\eta} + \frac{\partial \Upsilon_{i+1}}{\partial \theta_i}\frac{\mathrm{d}\theta_i}{\mathrm{d}\eta} + \frac{\partial \Upsilon_{i+1}}{\partial \upsilon_i}\frac{\mathrm{d}\upsilon_i}{\mathrm{d}\eta} + \frac{\partial \Upsilon_{i+1}}{\partial \eta}\\[4pt] &= \frac{\partial \Upsilon_{i+1}}{\partial \nabla L_i}\left(\frac{\partial^2 L}{\partial \theta_i^2}\frac{\mathrm{d}\theta_i}{\mathrm{d}\eta} + \frac{\partial^2 L}{\partial \eta \partial \theta_i}\right) + \frac{\partial \Upsilon_{i+1}}{\partial \theta_i}\frac{\mathrm{d}\theta_i}{\mathrm{d}\eta} + \frac{\partial \Upsilon_{i+1}}{\partial \upsilon_i}\frac{\mathrm{d}\upsilon_i}{\mathrm{d}\eta} + \frac{\partial \Upsilon_{i+1}}{\partial \eta}\\[4pt] &= \left(\frac{\partial \Upsilon_{i+1}}{\partial \theta_i} + \frac{\partial \Upsilon_{i+1}}{\partial \nabla L_i}\frac{\partial^2 L}{\partial \theta_i^2}\right)\frac{\mathrm{d}\theta_i}{\mathrm{d}\eta} + \frac{\partial \Upsilon_{i+1}}{\partial \upsilon_i}\frac{\mathrm{d}\upsilon_i}{\mathrm{d}\eta} + \left(\frac{\partial \Upsilon_{i+1}}{\partial \nabla L_i}\frac{\partial^2 L}{\partial \eta \partial \theta_i} + \frac{\partial \Upsilon_{i+1}}{\partial \eta}\right)\end{aligned}$$

.

Rewriting this in the block-matrix form results in

$$\begin{pmatrix}\frac{\mathrm{d}\theta_{i+1}}{\mathrm{d}\eta}\\[4pt]\frac{\mathrm{d}\upsilon_{i+1}}{\mathrm{d}\eta}\end{pmatrix} = \begin{pmatrix}\frac{\partial \Upsilon_{i+1}}{\partial \theta_i} + \frac{\partial \Upsilon_{i+1}}{\partial \nabla L_i}\frac{\partial^2 L}{\partial \theta_i^2} & \frac{\partial \Upsilon_{i+1}}{\partial \upsilon_i}\end{pmatrix}\begin{pmatrix}\frac{\mathrm{d}\theta_i}{\mathrm{d}\eta}\\[4pt]\frac{\mathrm{d}\upsilon_i}{\mathrm{d}\eta}\end{pmatrix} + \frac{\partial \Upsilon_{i+1}}{\partial \nabla L_i}\frac{\partial^2 L}{\partial \eta \partial \theta_i} + \frac{\partial \Upsilon_{i+1}}{\partial \eta}. \tag{15}$$

Or, alternatively,

$$\frac{\mathrm{d}\Upsilon_{i+1}}{\mathrm{d}\eta} = \left(\underbrace{\frac{\partial \Upsilon_{i+1}}{\partial \theta_i} + \frac{\partial \Upsilon_{i+1}}{\partial \nabla L_i}\frac{\partial^2 L}{\partial \theta_i^2}}_{(P+O)\times P} \quad \overbrace{\frac{\partial \Upsilon_{i+1}}{\partial \upsilon_i}}^{(P+O)\times O}\right)\underbrace{\frac{\mathrm{d}\Upsilon_i}{\mathrm{d}\eta}}_{(P+O)\times M} + \frac{\partial \Upsilon_{i+1}}{\partial \nabla L_i}\frac{\partial^2 L}{\partial \eta \partial \theta_i} + \frac{\partial \Upsilon_{i+1}}{\partial \eta}, \tag{16}$$

where $P$, $O$, and $M$ are the sizes of $\theta$, $\upsilon$, and $\eta$ correspondingly.

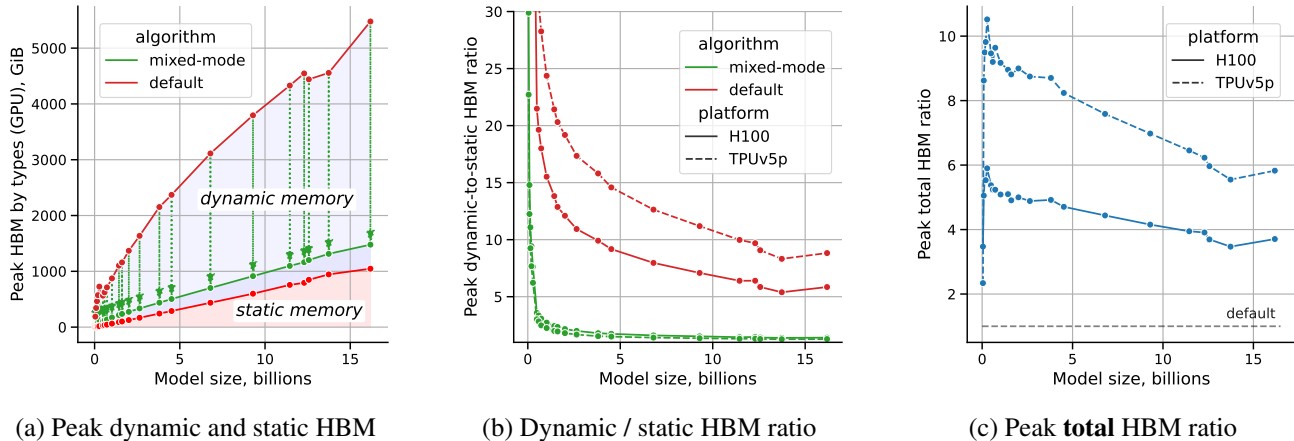

(a) Peak dynamic and static HBM     (b) Dynamic / static HBM ratio     (c) Peak **total** HBM ratio

*Figure 8.* Static and dynamic peak HBM w.r.t. model size.

Assuming that the function approximator and loss function have continuous second-order derivatives, which is typically the case for neural networks, the following identities hold (c.f. *Schwarz's theorem*):

$$\frac{\partial^2 L}{\partial \theta_i^2}^T = \frac{\partial^2 L}{\partial \theta_i^2}, \quad \frac{\partial^2 L}{\partial \eta \partial \theta_i}^T = \frac{\partial^2 L}{\partial \theta_i \partial \eta}.$$

Combining them with Equation (16) we can rewrite the vector-by-hessian (VHP) and vector-by-mixed-derivatives-matrix (VMP) products into their transposed versions, i.e. hessian-by-vector (HVP) and mixed-derivatives-matrix-by-vector (MVP) products:

$$\underbrace{v \frac{\partial \Upsilon_{i+1}}{\partial \nabla L_i} \frac{\partial^2 L}{\partial \theta_i^2}}_{\textbf{inefficient VHP}} = \left( \frac{\partial^2 L}{\partial \theta_i^2}^T \left( v \frac{\partial \Upsilon_{i+1}}{\partial \nabla L_i} \right)^T \right)^T = \left\{ \text{since } \frac{\partial^2 L}{\partial \theta_i^2}^T = \frac{\partial^2 L}{\partial \theta_i^2} \right\} = \left( \underbrace{\frac{\partial^2 L}{\partial \theta_i^2} \overbrace{\left( v \frac{\partial \Upsilon_{i+1}}{\partial \nabla L_i} \right)^T}^{\textit{normal VJP}}}_{\textbf{efficient HVP}} \right)^T$$

$$\underbrace{v \frac{\partial \Upsilon_{i+1}}{\partial \nabla L_i} \frac{\partial^2 L}{\partial \eta \partial \theta_i}}_{\textbf{inefficient VMP}} = \left( \frac{\partial^2 L}{\partial \eta \partial \theta_i}^T \left( v \frac{\partial \Upsilon_{i+1}}{\partial \nabla L_i} \right)^T \right)^T = \left\{ \text{since } \frac{\partial^2 L}{\partial \eta \partial \theta_i}^T = \frac{\partial^2 L}{\partial \theta_i \partial \eta} \right\} = \left( \underbrace{\frac{\partial^2 L}{\partial \theta_i \partial \eta} \overbrace{\left( v \frac{\partial \Upsilon_{i+1}}{\partial \nabla L_i} \right)^T}^{\textit{normal VJP}}}_{\textbf{efficient MVP}} \right)^T.$$

## A.3. Handling static device memory

In the terminology introduced in Section 4, static device memory is used for storing inputs and outputs, parameters $\theta$, states $\upsilon$, and checkpointed gradient and allocated at the beginning of the on-device computation for its whole lifetime. So far, the reported performance metrics reflected only changes in dynamic memory usage because *MixFlow-MG* operates on per-inner step basis, i.e. it does not change the static memory allocations.

Figure 8(a) shows dynamic and static memory distribution for the chinchilla scaling experiments from Section 5.3. As can be seen, *MixFlow-MG* reduces dynamic memory by 10-25x, thus turning static memory into the dominating part of the allocated device memory. This gets exacerbated by the fact that, as models and their optimizers' states become larger, the overall dynamic-to-static ratio shrinks from 50-100 to 5-10 for default implementation, as depicted in Figure 8(b). In total, this causes peak HBM memory gains to reduce from 10-25x (Figure 7) to 4-6x (Figure 8(c)).

Fortunately, the static memory factor can be reduced by several folds with the following techniques or their combinations:

- For distributed setups with $D$ interconnected devices, static tensors can be efficiently (i.e. with minimal communication overhead) distributed using Fully-Sharded Data Parallelism (FSDP) (Rajbhandari et al., 2020), thus reducing the static memory allocation per device by $D$ times.

- For momentum-based inner-level optimisers, such as Adam (Kingma, 2014), one can use the technique proposed in Maclaurin et al. (2015) to invert per-step updates during backward pass instead of storing them in static device memory; moreover, combining it with per-inner update remat from Section 4 can allow to avoid computational overheads; the same principle holds for arbitrary optimisers applied to the class of reversible networks (Behrmann et al., 2019; Kitaev et al., 2020; Mangalam et al., 2022).

- The default per-inner update rematerialisation strategy can be improved using dynamic programming (Griewank, 1992), allowing to reduce static memory usage from linear to logarithmic by $T$ (the number of inner updates per each outer update).

All of these techniques are fully compatible with *MixFlow-MG* and allow to achieve the 10-25x gains shown in Figure 7 with affordable (if not zero) compute overhead. We leave the implementation details of these methods outside the scope of this paper, as they can be found in the corresponding original works.

### A.4. Python code for mixed-mode bilevel gradients in JAX and PyTorch

*Code Listing 1.* JAX implementation for *fwdrev_grad* in Algorithm 2

```python
def get_fwdrev_grad_fn(inner_loss_fn):
  """Returns a function implementing `grad(inner_loss_fn)`.

  The returned function has a custom-defined VJP rule for implementing
  forward-over-reverse mode for Hessian-by-vector products that emerge in
  the meta gradient / bilevel optimisation scenario. This custom rule can save
  a substantial amount of memory and compute compared with default JAX autodiff.

  Args:
    inner_loss_fn: a function implementing inner loss calculation. It must
      accept `params` as the first argument.

  Returns:
    A function implementing `grad(inner_loss_fn)` with a custom-defined VJP
    rule for forward-over-reverse Hessian-by-vector products.
  """

  @jax.custom_vjp
  def fwdrev_grad_fn(params, *inputs):
    """Pure implementation."""
    return jax.grad(inner_loss_fn)(params, *inputs)

  def fwdrev_grad_fn_forward_pass(params, *inputs):
    """Forward pass implementation."""
    return fwdrev_grad_fn(params, *inputs), (params, inputs)

  def fwdrev_grad_fn_backward_pass(residuals, ct):
    """Backward pass implementation."""
    (params, inputs) = residuals
    grad_loss_fn = jax.grad(inner_loss_fn, argnums=range(len(inputs) + 1))
    _, hvp_ct = jax.jvp(lambda p: grad_loss_fn(p, *inputs), (params,), (ct,))
    return hvp_ct

  fwdrev_grad_fn.defvjp(
      fwdrev_grad_fn_forward_pass, fwdrev_grad_fn_backward_pass
  )

  return fwdrev_grad_fn
```

*Code Listing 2.* PyTorch implementation for *fwdrev_grad* in Algorithm 2

```python
def get_fwdrev_grad_fn(inner_loss_fn):
  """Returns a function implementing 'grad(inner_loss_fn)'.

  The returned function has a custom-defined VJP rule for implementing
  forward-over-reverse mode for Hessian-by-vector products that emerge in
  the meta gradient / bilevel optimisation scenario. This custom rule can save
  a substantial amount of memory \& compute compared with default implementation.

  Args:
    inner_loss_fn: a function implementing inner loss calculation. It must
      accept 'params' as the first argument.

  Returns:
    A function implementing 'grad(inner_loss_fn)' with a custom-defined VJP
    rule for forward-over-reverse Hessian-by-vector products.
  """

  class FwdRevGrad(torch.autograd.Function):

    @staticmethod
    def forward(context, params, *inputs):
      """Forward pass implementation."""
      context.save_for_backward(params, *inputs)
      return torch.func.grad(inner_loss_fn)(params, *inputs)

    @staticmethod
    def backward(context, ct):
      """Backward pass implementation."""
      params, *inputs = context.saved_tensors
      grad_loss_fn = torch.func.grad(loss, argnums=tuple(range(len(inputs) + 1)))
      _, hvp_ct = torch.func.jvp(lambda p: grad_loss_fn(p, *inputs), (params,), (ct,))
      return hvp_ct

  return FwdRevGrad.apply
```

## A.5. Python snippet for per-inner step gradient checkpointing with saving inner gradients

*Code Listing 3.* Python snippet for optimisations in Section 4

```python
def inner_step(...):  # Implements one inner step.
  d_params = grad_fn(params, inputs)
  d_params = jax.tree.map(
    lambda x: jax.ad_checkpoint.checkpoint_name(x, 'inner_grads'), d_params
  )
  ...

def outer_step(...):  # Implements the outer step.
  inner_step = jax.checkpoint(
    inner_step, policy=jax.checkpoint_policies.save_only_these_names('inner_grads'))
  new_params, ... = jax.lax.scan(inner_step, ...)
  ...
```

## A.6. Python code for the motivating example

*Code Listing 4.* Python implementation for the motivating example

```python
def get_toy_task(seed, B, M, T, D, use_loop_fusion, use_mixed_mode):
  """Returns a toy task and example arguments.

  Args:
    seed: a random seed.
    B: a batch size.
    M: a number of inner steps.
    T: a number of inner updates.
    D: a data and inner model size.
    use_loop_fusion: whether to use loop fusion.
    use_mixed_mode: whether to use mixed mode.

  Returns:
    A jitted function with arguments that correspond to the toy task.
  """
  rng1, rng2, rng3 = jax.random.split(jax.random.PRNGKey(seed), 3)
  params = jax.random.normal(rng1, (D, D))
  xs, targets = jax.random.normal(rng2, (2, T, B, D))
  val_x, val_target = jax.random.normal(rng3, (2, B, D))

  def toy_task(params, xs, targets, val_x, val_target):

    def apply(params, x):
      y = jnp.matmul(x, params)

      def f(y, i):
        return i * (2 + jnp.sin(y)) ** jnp.cos(y), ()

      if use_loop_fusion:
        for i in range(1, M + 1):
          y, _ = f(y, i)
      else:
        y, _ = jax.lax.scan(f, y, jnp.arange(1, M + 1))
      return y

    def loss(params, x, target):
      return jnp.mean((apply(params, x) - target) ** 2)

    def meta_loss(params):
      if use_mixed_mode:
        grad_fn = get_fwdrev_grad_fn(loss)
      else:
        grad_fn = jax.grad(loss)

      def inner_step(params, x_and_target):
        d_params = grad_fn(params, *x_and_target)
        params = jax.tree.map(lambda p, dp: p - 1e-3 * dp, params, d_params)
        return params, ()

      params, _ = jax.lax.scan(inner_step, params, (xs, targets))
      return loss(params, val_x, val_target)

    meta_grad = jax.grad(meta_loss)(params)
    return meta_grad

  return jax.jit(toy_task), (params, xs, targets, val_x, val_target)
```

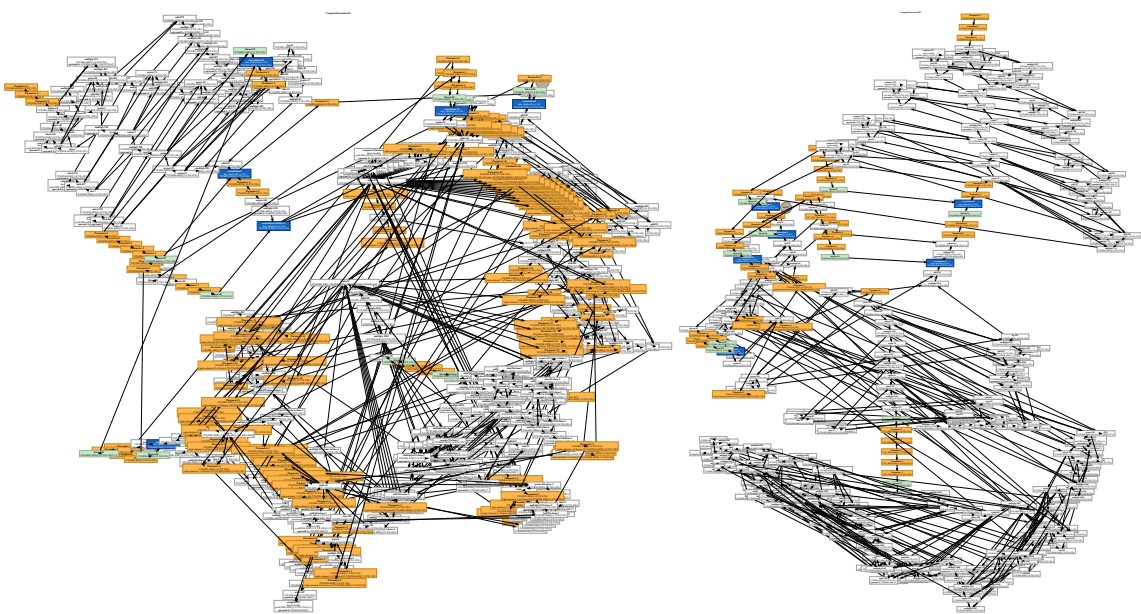

*Figure 9.* HLO graph for the motivational example. Data nodes are depicted in orange, compute operations (multiplications, trigonometric functions etc.) in gray. It can be seen that the mixed-mode version contains far fewer data blocks. Also, this example demonstrates the complexity of the underlying low-level programs and the huge role of compiler in optimising raw computational graphs.

## A.7. Compiled computational graphs for the motivating example

See Figure 9.

## A.8. Detailed ablations on all used optimisations

See Figure 10 and Table 2 for ablations on 489M model and Table 3 for step time measurements on 44M model, which fits into single-core device memory.

*Table 2.* Case study for 489M transformer.

| Optimisations | | | 489M transformer | | | |
|---|---|---|---|---|---|---|
| Mixed mode | Block remat | Save grads | GPU | | TPU | |
| | | | HBM (G) | Time (s) | HBM (G) | Time (s) |
| - | - | - | 371.2 | N/A | 273.9 | N/A |
| - | - | + | 363.7 | N/A | 176.6 | N/A |
| - | + | - | 180.1 | N/A | 123.7 | N/A |
| - | + | + | 182.4 | N/A | 130.8 | N/A |
| + | - | - | 286.0 | N/A | 168.1 | N/A |
| + | - | + | 289.2 | N/A | 176.8 | N/A |
| + | + | - | 174.8 | N/A | **43.8** | 5.13 |
| + | + | + | **54.8** | **5.45** | 46.9 | **4.12** |

## A.9. Sweeps over data regimes for TPUs

See Figure 11.

## A.10. Models and hyperparameters

For sweeps in Figure 5 we used hyperparameters from Table 4. For benchmarks in Figure 6 we used the models from Table 5. For scaling plots in Figure 7 and Figure 8 we used models from Table 6 with batch size 4 and 2 inner steps per outer update.

*Table 3.* Case study for 44M transformer.

| Optimisations | | | 44M transformer | | | |
|---|---|---|---|---|---|---|
| Mixed mode | Block remat | Save grads | GPU | | TPU | |
| | | | HBM (G) | Time (s) | HBM (G) | Time (s) |
| - | - | - | 94.2 | N/A | 70.2 | 0.75 |
| - | - | + | 76.6 | N/A | 45.8 | **0.70** |
| - | + | - | 54.2 | 1.33 | 32.8 | 1.05 |
| - | + | + | 54.5 | 1.30 | 34.1 | 1.03 |
| + | - | - | 76.4 | N/A | 45.1 | 0.88 |
| + | - | + | 76.6 | N/A | 45.5 | **0.70** |
| + | + | - | 45.2 | 1.51 | **12.7** | 1.17 |
| + | + | + | **16.4** | **1.19** | **12.9** | 0.94 |

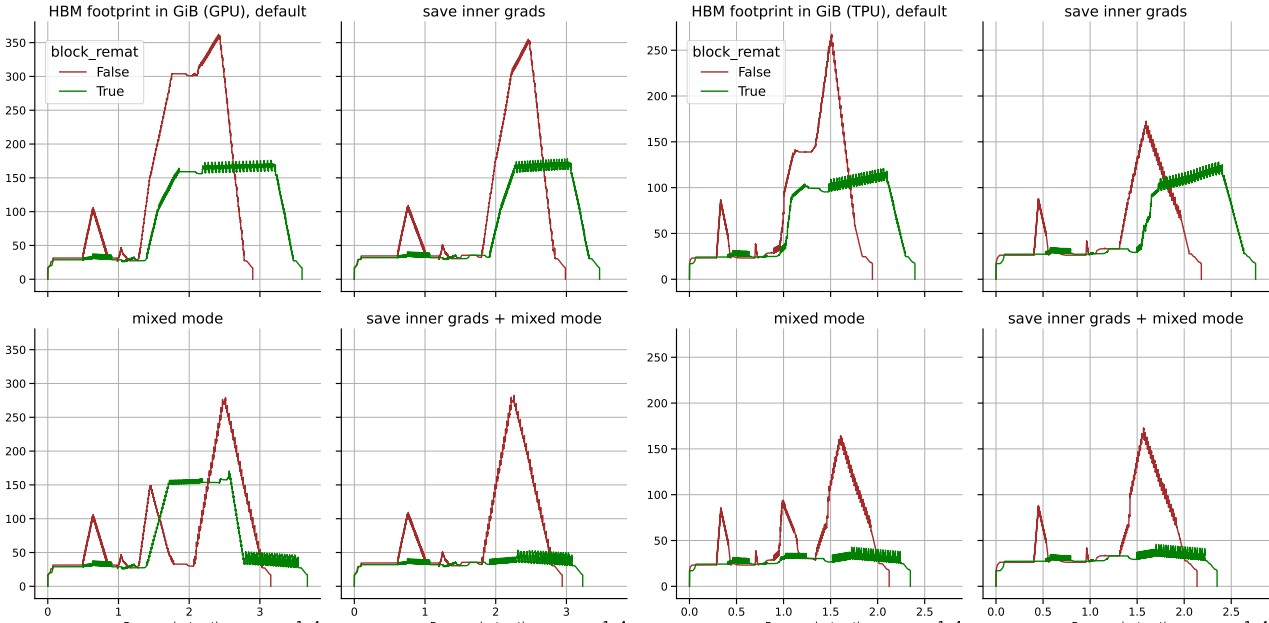

*Figure 10.* All combinations of the used optimisation from Section 4 for 489M model. Note that GPU required saving inner gradients for peak memory gains, while TPU needs it only for reducing step time. Mixed-mode differentiation and model blocks' rematerialisations are critical for both cases.

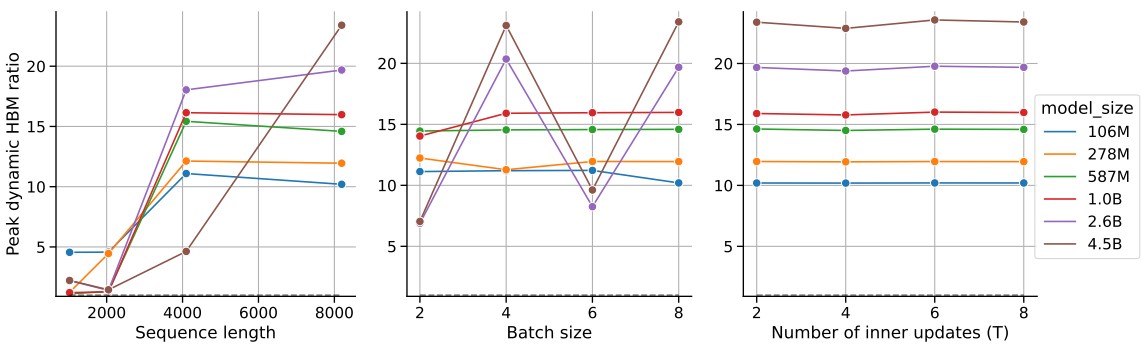

*Figure 11.* Sweep over data regimes for chinchilla models for TPUs. For GPUs see Figure 5. The results are more noisy due to TPU-specific optimisations and memory layout (e.g. memory padding).

*Table 4.* Sweep over data regimes in Figure 5: hyperparameters, values, and descriptions. When plotting each of three per-axis plots, we used the maximum values for the other two axes (e.g. for the sequence length plot we used batches of size 8 with 8 inner updates).

| Parameter | Values | Description |
|---|---|---|
| Model size ($\times 10^6$) | {106, 278, 587, 1018, 2639, 4516} | Parameters in inner transformer |
| # of inner updates ($T$) | {2, 4, 6, 8} | Inner updates per outer update |
| Batch size | {2, 4, 6, 8} | Inner model's batch size |
| Sequence length | {1024, 2048, 4096, 8192} | Context length |

*Table 5.* Chinchilla models used in sweeps over each of the components, Section 5.3

| Sweep over | d_model | ffw_size | kv_size | n_heads | n_layers |
|---|---|---|---|---|---|
| d_model | 128-2048 | 1024 | 16-256 | 8 | 16 |
| ffw_size | 384 | 512-8192 | 32 | 8 | 16 |
| n_heads | 768 | 1024 | 24-384 | 2-32 | 16 |
| n_layers | 256 | 1024 | 32 | 8 | 4-64 |

*Table 6.* Chinchilla models from Hoffmann et al. (2022a) used in scaling benchmarks, Section 5.3.

| Parameters (million) | d_model | ffw_size | kv_size | n_heads | n_layers |
|---|---|---|---|---|---|
| 44 | 512 | 2048 | 64 | 8 | 8 |
| 90 | 640 | 2560 | 64 | 10 | 13 |
| 140 | 768 | 3072 | 64 | 12 | 15 |
| 196 | 896 | 3584 | 64 | 14 | 16 |
| 278 | 1024 | 4096 | 64 | 16 | 18 |
| 489 | 1280 | 5120 | 128 | 10 | 21 |
| 587 | 1408 | 5632 | 128 | 11 | 21 |
| 724 | 1536 | 6144 | 128 | 12 | 22 |
| 1,018 | 1792 | 7168 | 128 | 14 | 23 |
| 1,429 | 2048 | 8192 | 128 | 16 | 25 |
| 1,609 | 2176 | 8704 | 128 | 17 | 25 |
| 2,007 | 2304 | 9216 | 128 | 18 | 28 |
| 2,639 | 2560 | 10240 | 128 | 20 | 30 |
| 3,802 | 2816 | 11264 | 128 | 22 | 36 |
| 4,516 | 3072 | 12288 | 128 | 24 | 36 |
| 6,796 | 3584 | 14336 | 128 | 28 | 40 |
| 9,293 | 4096 | 16384 | 128 | 32 | 42 |
| 11,452 | 4352 | 17408 | 128 | 32 | 47 |
| 12,295 | 4608 | 18432 | 128 | 36 | 44 |
| 12,569 | 4608 | 18432 | 128 | 32 | 47 |
| 13,735 | 4864 | 19456 | 128 | 32 | 47 |
| 16,183 | 5120 | 20480 | 128 | 40 | 47 |

