# OpenReview forum: "Scalable Meta-Learning via Mixed-Mode Differentiation"
_ICML.cc/2025/Conference — ICML 2025 poster_

### Official Review · Reviewer_zFod · 2025-02-24

**Overall Recommendation:** 4

**Summary:**

This paper develops a novel efficient bi-level optimization framework, called MixFlow-MG. Based on the fact that, when computing HVPs, forward-over-reverse mode is usually much more efficient than other modes, the authors first reparameterize the update function, and replaces the usual VHP and VMP computations in the meta-gradient into HVP and MVP computations, respectively, significantly improving the computational efficiency of meta-gradient computation. The analysis show that the proposed method, combined with other techniques such as block rematerialization and inner-gradient saving, significantly reduce the memory overhead, as well as wall-clock speedups on various learning scenarios, making meta-learning feasible even on extremely large models with billions of parameters.

**Claims And Evidence:**

Overall, I think this paper is very strong. As a meta-learning researcher, it's really interesting and exciting to see that bi-level optimization now become scalable (up to some degree) even with LLMs. All the claims about its computational efficiency are strongly supported by clear evidence, everywhere. All the figures, from Figure 1 to Figure 7, clearly demonstrate that the proposed method dramatically reduce the memory burden and wall time, in a systematic way. I didn't find any issue regarding missing evidence for this paper.

**Essential References Not Discussed:**

I didn't find any missing references.

**Experimental Designs Or Analyses:**

Again, while I'm not an expert in low-level implementation and relevant terminologies, I can clearly see that the analysis are systematic and well controlled. I don't find any relevant issues.

**Methods And Evaluation Criteria:**

The propose method is quite simple - just reparameterize the update function, and switch the order of computation to change VHP and VMP into more efficient HVP and MVP.

Honestly, I'm not super expert in low-level implementation, so I was not aware that HVP can be more efficiently computed with the forward-over-reverse mode. While I still don't fully understand the technical details as to why this is the case, but anyway, assuming this is true, I believe the proposed method is very reasonable and is a big contribution to the meta-learning field.

I don't find any specific issue regarding the evaluation criteria.

**Other Comments Or Suggestions:**

Just in case, what if you add a simple experiment (in the appendix) showing that the proposed algorithm produces the exact meta-gradient? I don't think it's essential, but it might be helpful to make readers more strongly believe the correctness of the proposed algorithm.

**Other Strengths And Weaknesses:**

I don't see any weaknesses of this paper.

For the strengths, see above.

**Questions For Authors:**

I have a few simple questions. I'm open to further increase my score if those are resolved.

1. I didn't fully understand how precisely the backprop is done with your algorithm. The one-step unrolling equation in Eq.(6) is for deriving the FMD algorithm, as far as I know (I might be wrong). But it seems that the proposed algorithm seems to be a kind of backprop algorithm. Could you elaborate on this? It's not entirely clear in Algorithm 1 and 2. Specifically, what is "fwdrev_grad"? Is it something you implemented, or an existing API? (I'm not familiar with JAX, so was not fully understand A.4)

2. Regarding the proposed reparameterization, I didn't know that the publicly available auto-diff frameworks implements backprop in such a way (because there are extra terms in Eq.(6) compared to when we do not reparameterize). I thought they follow the original parameterization with $\Phi$ (and in this case, the proposed trick cannot be applied as the blue and red terms in Eq.(6) disappear). Could you provide more details or references about this?

3. It seems that most of the experiments assume rather short inner-learning trajectory, like $T \in \{2,4,8\}$. Could you show the efficiency of your method when $T$ is very large, like $T=10^3$ or $10^4$?

4. I'm a bit confused with the experiment in Section 3.2. Why is it a bi-level optimization problem? Could you elaborate more? What does it mean by controlling $M$?

**Relation To Broader Scientific Literature:**

This paper makes meta-learning scalable, so it can be a big contribution to the LLM and other relevant fields as well, i.e., all the area where meta-learning was infeasible due to the excessive memory issue.

**Theoretical Claims:**

I carefully checked the derivation of meta-gradient in Section A.2, and it is correct.

---

> ### Author Rebuttal · Authors · 2025-04-01
>
> We thank the Reviewer for the thorough feedback and for pointing out the parts of the paper that weren’y sufficiently clear. We will incorporate out responses below into the final version to improve clarity.
>
> > I was not aware that HVP can be more efficiently computed with the forward-over-reverse mode…I still don't fully understand the technical details
>
> Using FMD for HVPs avoids having to store some intermediate activations (which standard RMD uses). This yields the exact same gradient, but often with much less memory and compute spent, especially for the case of gradient checkpointing (Sections 2.2, 3, A.1). We will refine the narrative in the final version to consolidate all the details in one place.
>
> > Just in case, what if you add a simple experiment (in the appendix) showing that the proposed algorithm produces the exact meta-gradient?
>
> The proposed algorithm is theoretically identical (up to machine precision) to the well-studied  BPTT methods. Assuming that JAX and PyTorch correctly compute meta-gradients, we double-check this by validating that all our MixFlowMG runs produce the same numerical results as the baselines, with up to 1e-6 absolute and 1e-4 relative errors, which are common for 32-bit floating point arithmetic.
>
> **Q1**
>
> Indeed, Eq.(6) reveals *the terms (VHPs and VMPs) that are calculated using FMD* in the proposed algorithm. The crucial part here is that *the rest of the terms are still calculated using backprop , or RMD*, which makes *MixFlowMG a mixed-mode differentiation method* (hence the name – Mixed-Mode Meta-Gradients).
>
> `fwdrev_grad` in Alg.(2) is the result of calling `get_fwdrev_grad_fn` in f A.4: it’s a function for calculating the gradient of inner loss $L$ w.r.t. inner parameters $\theta$, but with custom differentiation rules that constitute MixFlowMG. These rules instruct JAX to compute VHPs and VMPs as more optimal HVPs and MVPs – this is why MixFlowMG can be implemented in just a few lines of code, yet yielding substantial performance wins in practice.  Precisely the same mechanism is used in the Pytorch implementation for MixFlowMG, which we will add to the final version of the paper.
>
> **Q2**
>
> To the best of our knowledge, highly-rated publicly available libraries for meta-learning (e.g. [1], [2]), indeed, use the parameterisation from Eq.(3) and implement meta-gradients as outlined in Alg.(1).
>
> In our paper, we demonstrate that by introducing a simple reparameterisation Eq.(4) and using a slightly modified yet theoretically equivalent Alg.(2), those libraries can easily adopt MixFlowMG and benefit from significant compute and memory gains. To re-emphasize, Eq.(3) + Alg.(1) is theoretically equivalent to Eq.(4)+Alg.(2), as shown in the paper.
>
> Essentially, the proposed re-parameterisation in Eq.(4) allows for revealing the blue and red terms in Eq.(6), which are implicitly calculated using RMD in the autodiff frameworks. Explicitly surfacing these terms in the update equations allows for calculating them via FMD, which is the core idea of MixFlowMG. Hope this clarifies the contribution of the paper!
>
> [1] Arnold, Sébastien MR, et al. "learn2learn: A library for meta-learning research." arXiv preprint arXiv:2008.12284 (2020), https://github.com/learnables/learn2learn
>
> [2] Metz, Luke, et al. "Practical tradeoffs between memory, compute, and performance in learned optimizers." Conference on Lifelong Learning Agents. PMLR, 2022, https://github.com/google/learned_optimization
>
> **Q3**
>
> Yes, this is possible. The primary goal of MixFlowMG is to reduce memory footprint per single inner step (i.e. the dynamically allocated memory), while increasing T demands more statically allocated memory (Section 4 and Figure 2). That being said, dynamic memory gains are directly convertible to extra static memory: for example, if the size of parameters and states is $O(P)$, and the dynamic memory reduction thanks to MixFlowMG is of order $O(WP)$, then T can be scaled as $O(W)$ in a naive way. Typically, for smaller $P$ and/or higher compute intensity per parameter this win factor is higher, as e.g. in Section 3.2.
>
> We described more efficient ways to scale static memory in A.3, all of which are fully compatible with MixFlowMG. Those include distributing static memory over $d$ devices, effectively reducing it by $d$ times, which worked for us.
>
> **Q4**
>
> The setup from Section 3.2 is essentially MAML: it is a BLO because the outer parameters $\eta$ define the initialisation for the inner parameters $\theta_0 = \eta$; after updating $\theta$ for T steps, we calculate the validation loss on $\theta_T$ and propagate the gradients for $\partial V / \partial \eta$ back to $\theta_0$, to perform one outer update.
>
> Parameter $M$ controls the size of a single inner update in this toy example Eq.(9): for larger $M$, one inner step requires storing more intermediate activations for RMD. Varying $M$ in Figure 1 demonstrates how the gains from using MixflowMG scale with the size of the inner updates.

---

> > ### Comment · Reviewer_zFod · 2025-04-04
> >
> > Thanks for the clarification. They helped me to understand this paper better.
> >
> > On the other hand, I somewhat agree with the Reviewer zzTN that this paper should be compared with the approximate BLO methods, such as Neumann IFT by Lorraine et al. and "Making scalable meta learning practical" paper by Choe et al.
> >
> > I initially thought that this paper is not directly comparable to those approximations because this paper is about computing the exact meta-gradients. However, now I think that the benefit of computing the exact meta-gradient should be demonstrated somehow. This is because, in some cases, the exact meta-gradients suffer from vanishing/exploding gradient issue, especially when the length of inner-learning trajectory T is long (see https://openreview.net/pdf?id=5lko4-9cRAk). That being said, the exact meta-gradients are not always better than those approximations, and it's still questionable whether to use the proposed MixFlowMG or those approximations, for the practitioners.
> >
> > For now, I maintain my current score, but I'd like to encourage the authors to compare the proposed MixFlowMG against those approximations and analyze when the proposed method outperforms and when it does not, in terms of the wall-clock meta-convergence (assuming that the memory footprints are similar across the methods).
> >
> > If it's done corretly, I would be happy to champion this paper to other reviewers and the AC.

---

> > > ### Author Response · Authors · 2025-04-06
> > >
> > > We really appreciate the engagement with this work and the desire to put it further in context wrt approximate BLO methods.
> > >
> > > We carefully studied the cited works and main assumptions of approximate methods based on the Implicit Function Theorem (i.e. stationarity and regularity conditions for the inner optimisation, as discussed in Section 2 in [1]; additional assumptions Eq. (3) and (5) in [2]), as well as the implied theoretical limitations (as discussed in e.g. Section 3.1 from [3]). While we acknowledge that the practical importance of these assumptions can vary by application and agree that more evidence supporting scaling exact meta-gradients for BLO would be valuable, making an empirical comparison with the cited works will be infeasible for us in the rebuttal period.
> > >
> > > Instead, we urge the reviewers to consider the significant body of literature pursuing meta-gradient methods for BLO in various contexts, including recent works using LLMs which are certainly gated by memory considerations ([3] only use 82M parameter models in meta-training).
> > >
> > > While long-T meta-gradients have known issues, our work can allow the use of an order of magnitude larger models even for shorter horizons and will enable further scale of research into the pathologies of long-horizon meta-optimization (and thereby possible mitigations to the issues found there). For example, meta-training in [3] can be scaled up to 7x larger inner models with bigger batch sizes and longer Ts with MixFlowMG, as demonstrated in Table 2 of the paper.
> > >
> > > Overall, we are confident that this field of research is sufficiently active to warrant the broader adoption of our method, even as other BLO methods are studied further in parallel. We hope the reviewers agree that the quality of our specific contribution, its simple practical implementation, and the depth of our analysis are sufficient to enable other researchers to build forward from.
> > >
> > >
> > > [1] Lorraine, Jonathan, Paul Vicol, and David Duvenaud. "Optimizing millions of hyperparameters by implicit differentiation." International conference on artificial intelligence and statistics. PMLR, 2020.
> > >
> > > [2] Choe, Sang, et al. "Making scalable meta learning practical." Advances in neural information processing systems 36 (2023): 26271-26290.
> > >
> > > [3] Loo, Noel, et al. "Dataset distillation with convexified implicit gradients." International Conference on Machine Learning. PMLR, 2023.
> > >
> > > [4] Hu, Nathan Zixia, et al. "Meta-Learning Online Adaptation of Language Models." The 2023 Conference on Empirical Methods in Natural Language Processing.

---

### Official Review · Reviewer_zzTN · 2025-03-10

**Overall Recommendation:** 2

**Summary:**

The paper introduces MixFlow-MG, a method for scalable bilevel optimization in meta-learning using mixed-mode automatic differentiation. The core idea involves reparameterizing the inner-loop dynamics to replace inefficient reverse-over-reverse differentiation with forward-over-reverse mode. This reduces computational costs and memory usage. Experiments demonstrate up to 10× memory savings and 25% faster wall-clock time across tasks like hyperparameter learning, MAML, and loss weighting. The method is validated on large models (up to 16B parameters) and hardware (GPUs/TPUs), showing consistent gains in dynamic memory and compute efficiency.

**Claims And Evidence:**

1. The submission claims that the reverse-mode differentiation performed by the default autodiff can be suboptimal in practice. However, the underlying reasons for this drawback are not clearly explained in the submission.

2. Equations (7) and (8) claim that the transport versions of VHP and VMP lead to more efficient computation; however, this is not clearly explained.

**Essential References Not Discussed:**

The implicit gradient-based algorithms are not discussed in the paper.

[1] Lorraine, Jonathan, Paul Vicol, and David Duvenaud. "Optimizing millions of hyperparameters by implicit differentiation." International conference on artificial intelligence and statistics. PMLR, 2020.

[2] Rajeswaran, Aravind, et al. "Meta-learning with implicit gradients." Advances in neural information processing systems 32 (2019).

[3] Gao, Boyan, et al. "Loss function learning for domain generalization by implicit gradient." International Conference on Machine Learning. PMLR, 2022.

**Experimental Designs Or Analyses:**

Only three bilevel optimization-based methods are evaluated using the proposed hypergradient computation strategy.

Lack of comparison with the recent methods focusing on LLM for instance [1].

[1] Choe, Sang, et al. "Making scalable meta learning practical." Advances in neural information processing systems 36 (2023): 26271-26290.

**Methods And Evaluation Criteria:**

The papers evaluate and compare the memory overhead introduced by the proposed algorithm and various baseline methods across different meta-learning algorithms. However, the performance of the bilevel optimization-based methods is not reported.

**Other Comments Or Suggestions:**

See the above sections.

**Other Strengths And Weaknesses:**

The main deep learning and meta-learning implementation trend is still PyTorch. However, this submission only demonstrates the algorithm on the VJP and JVP-friendly platform, JAX.

**Questions For Authors:**

See the above sections.

**Relation To Broader Scientific Literature:**

The submission focuses on efficient hypergradient computation in bilevel optimization. I believe there is no clear connection to broader scientific research beyond the scope of meta-learning.

**Theoretical Claims:**

There are no theorems proposed in the submission.

---

> ### Author Rebuttal · Authors · 2025-04-01
>
> We thank the Reviewer for the actionable feedback. We add clarifications and respond to the questions below.
>
> > The submission claims that the reverse-mode differentiation performed by the default autodiff can be suboptimal in practice. However, the underlying reasons for this drawback are not clearly explained … Equations (7) and (8) claim that the transport versions of VHP and VMP lead to more efficient computation; however, this is not clearly explained.
>
> Just to be sure, we would like to clarify that the claim about inefficiency of the reverse-mode differentiation is only related to the bilevel gradient optimisation (BLO) setups. We thank the Reviewer for pointing out that this wasn't sufficiently clear.
>
> Essentially, the transposing of the Hessian allows us to use forward-mode differentiation here, which avoids having to store intermediate activations (which standard reverse-mode differentiation uses). This results in the exact same gradient, but often with much less memory and compute spent, especially for the case of gradient checkpointing. The reason we can do it here is that the BPTT-based BLO methods always happen to end up with symmetric Hessian there, whose transpose is trivial.
>
> Although we discuss each of these parts in the paper (Sections 2.2, 3, A.1), we will refine the narrative in the final version to consolidate all the details in one place to improve clarity.
>
> >Only three bilevel optimization-based methods are evaluated using the proposed hypergradient computation strategy… Lack of comparison with the recent methods focusing on LLM…
>
> We agree that trying the proposed algorithm in more specific instances of BLO is an important direction. In particular, the combination with LLMs is especially interesting, because these setups already are often scaled all the way to what could possibly fit on device memory.
>
> In this paper, we showed that MixFlowMG yields exactly the same gradients (up to machine precision) as the existing Backpropagation-Through-Time (BPTT)-based algorithms in BLO, thus ensuring that their experimental results are fully transferable to MixFlowMG. Taking this into account, we decided to devote this paper to the practical aspects of MixFlowMG, i.e. its computational and memory benefits.
>
> Our method can be used to already reduce the computational requirements of existing highly performant BPTT-based approaches, as well as to unlock previously-infeasible setups which would otherwise have required too much memory. For such examples (scaling inner-loop length, LLMs), we’d like to kindly ask the Reviewer to have a look at our other responses due to the size limit.
>
> > The implicit gradient-based algorithms are not discussed in the paper.
>
> Implicit gradients are another BLO strategy with different pros and cons that are worth research in their own right and which we consider orthogonal to our specific contribution. For the readers who would like to learn more about the broader scope of the BLO, we included references to the broad surveys, e.g. [1], which have detailed sections on implicit gradients. We will add a separate paragraph mentioning alternatives to BPTT to the final version and include the provided references.
>
> >Equation (5) is not rigorous, as it sets the second derivative term to zero.
>
> Typically, the validation loss $V(\theta_T)$ does not depend on the internal optimiser’s state $\upsilon_T$, hence we set the mentioned derivative to zero in the corresponding equations. We will amend them to include this term in the most general form. Note that this does not affect the paper’s results and analysis, as all the derivations remain correct.
>
> > The submission focuses on efficient hypergradient computation in bilI believe there is no clear connection to broader scientific research beyond the scope of meta-learning.
>
> Our main focus is indeed BLO and efficient hypergradients. These methods are already impactful, e.g., in robotics, but we believe they will become increasingly important in AI research and applications, and so having efficient algorithms for this is important.
>
> That said, there are some surveys [1] which connect hypergradients with research areas beyond meta-learning (e.g. signal processing, adversarially robust training). MixFlowMG applies anywhere where a similar to meta-gradient structure appears, and similar computational gains can then be obtained there.
>
> >The main deep learning and meta-learning implementation trend is still PyTorch. However, this submission only demonstrates the algorithm on <...> JAX.
>
> We re-implemented the toy example in PyTorch and confirmed that the gains of MixFlowMG remain significant (up to 4x less memory on GPUs on this particular task). We will include this Pytorch implementation (~15 lines of code) in the final version.
>
> [1] Zhang, Yihua, et al. "An introduction to bilevel optimization: Foundations and applications in signal processing and machine learning." IEEE Signal Processing Magazine 41.1 (2024): 38-59

---

### Official Review · Reviewer_85Hz · 2025-03-13

**Overall Recommendation:** 3

**Summary:**

The paper introduces MixFlow-MG, an algorithm that uses mixed-mode differentiation for more efficient gradient-based bilevel optimization. It reduces memory consumption and computation time compared to standard methods.

**Claims And Evidence:**

Yes

**Essential References Not Discussed:**

No

**Experimental Designs Or Analyses:**

The authors validated their approach through:

Inefficiency analysis of standard Truncated-BPTT implementations.
Reparameterization to expose problem symmetry.
Demonstration of performance gains with mixed-mode differentiation.
Analysis of device memory footprint.
Evaluation of optimizations (block rematerialization, saving inner gradients).
Scaling analysis of dynamic memory requirements.

**Methods And Evaluation Criteria:**

Yes, the proposed methods and evaluation criteria are appropriate for addressing the problem of high computational costs in gradient-based bilevel optimization.

**Other Comments Or Suggestions:**

It is interesting that the proposed method enables the application of BLO to LLM. However, the paper does not show the concrete problem formulation that can be solved by the proposed method. The paper can be strengthened by including the experiment result on the application of BLO on LLM.

**Other Strengths And Weaknesses:**

Strengths

The paper provides a clear explanation of the problem and the proposed solution, and employs illustrative examples and visualizations to aid understanding.

The paper includes extensive experiments on the efficiency of the proposed method.

Weaknesses

While the experiments demonstrate the practical benefits of the proposed algorithm, the paper could be strengthened by including more experimental performance of the proposed method solving specific problems.

**Questions For Authors:**

In application of MAML, how many inner-level steps are applied? If it is one, what is the benefit of the proposed approach?

**Relation To Broader Scientific Literature:**

The paper proposes an efficient gradient computation for BLO.

**Theoretical Claims:**

N/A

---

> ### Author Rebuttal · Authors · 2025-03-31
>
> We thank the Reviewer for the thorough feedback and address the raised concerns below.
>
>
> > While the experiments demonstrate the practical benefits of the proposed algorithm, the paper could be strengthened by including more experimental performance of the proposed method solving specific problems.
> …  However, the paper does not show the concrete problem formulation that can be solved by the proposed method.
>
>
> We agree that trying the proposed algorithm in specific instances of BLO is an important direction. In this paper, we showed that MixFlowMG results in exactly the same gradients (up to machine precision, Section 3 and Appendix A.2.) as the existing Backpropagation-Through-Time (BPTT)-based algorithms, thus ensuring that their experimental results are fully transferable to MixFlowMG. Taking this into account, we decided to devote this paper to the practical aspects of MixFlowMG, i.e. its computational and memory benefits.
>
> We would like to highlight that the proposed algorithm can be combined with a wide variety of meta-learning algorithms, with the same guarantee of equivalence (up to numerical precision), as well as with future algorithms that have not yet been discovered.
>
> To share our enthusiasm about the experimental direction, we would like to highlight that several works, e.g. [1], [2], [3], reported positive effects of increasing the number of inner steps (up to 10 in [1], [2], and up to 100 in [3]) in the BPTT-based meta optimisation. Those results suggest that increasing this parameter by over 10 times with MixFlowMG is likely to follow this positive trend potentially leading to qualitatively new outcomes. That being said, to ensure the well-defined scope of the paper, we leave such investigations to future works and instead devote the experimental section mainly to the thorough technical analysis of the proposed algorithm.
>
>
> [1] _Figure 5_ in Finn, Chelsea, Pieter Abbeel, and Sergey Levine. "Model-agnostic meta-learning for fast adaptation of deep networks." International conference on machine learning. PMLR, 2017
>
> [2] _Table 2_ in Antoniou, Antreas, Harrison Edwards, and Amos Storkey. "How to train your MAML." arXiv preprint arXiv:1810.09502 (2018)
>
> [3] _Figure 2a_ in Rajeswaran, Aravind, et al. "Meta-learning with implicit gradients." Advances in neural information processing systems 32 (2019)
>
> > The paper can be strengthened by including the experiment result on the application of BLO on LLM.
>
> To the best of our knowledge, gradient-based bi-level optimisation for LLMs and especially practically strong BPTT-based algorithms still remain under-explored because of the prohibitive computational costs. Recently, it has become an emerging trend: for example, [1] uses meta-gradients with LLMs in the inner loop for the task of dynamic token-level loss reweighting at the fine-tuning stage. Despite using only 6 inner-loop steps, the authors reported significantly improved F1 scores on downstream QA tasks.
>
> That setup can be seamlessly scaled further with MixFlowMG, in terms of both model size and inner-loop length. We believe that its memory- and compute efficiency gains is a critical step towards the adoption of meta-gradients for LLM-related tasks. That is why, in the experimental part of the paper, we conduct a detailed analysis of the properties of MixFlowMG in common meta-learning workflows enhanced with real-world LLMs (the _Chinchilla_ scaling ladder).
>
> [1] Hu, Nathan Zixia, et al. "Meta-Learning Online Adaptation of Language Models." The 2023 Conference on Empirical Methods in Natural Language Processing.
>
> > In application of MAML, how many inner-level steps are applied? If it is one, what is the benefit of the proposed approach?
>
> We benchmarked the method on the number of inner-level steps T=2,4,6,8 for all tasks, including MAML. The results can be found in Figure 5(c), which shows that for the largest (4.5B) model, our method results in approximately a 10x reduction in memory requirements, and this relative gain stays fairly constant across different numbers of inner steps. We expand on this in Sections 4 and 5.3 of the paper (i.e. static and dynamic allocation patterns).
>
> We did observe significant memory and compute gains for T=1 as well, as 1-step hypergradients, despite not having vector-by-Hessian products (VHPs), still contain vector-by-mixed-derivatives-matrix products (VMPs) which are optimised in MixFlowMG (Proposition 3.1.). We decided to not include it in the plots because the corresponding points look like outliers compared with T>1, which may be confusing for the readers; we will comment on T=1 in the final version of the text.

---

### Official Review · Reviewer_ZDJP · 2025-03-16

**Overall Recommendation:** 3

**Summary:**

This paper proposes to smartly combine forward-mode and reverse-mode differentiations to efficiently perform JVP/HVP in bi-level optimization. This is motivated by the observation that different differentiation methods can show massively different performance in practice, despite their potentially same theoretical memory/compute complexity. The proposed MixFlow-MG achieves over 10x memory and up to 25% wall-clock time improvements over previous implementations of meta-learning algorithms in multiple benchmarks.

**Claims And Evidence:**

Yes.

**Essential References Not Discussed:**

N/A

**Experimental Designs Or Analyses:**

Given that the main focus of the paper is improving practical implementation efficiency of meta-learning algorithms, I appreciated that authors have provided extensive analyses on run-time and memory usage. The only issue I see is a lack of the meta learning performance analysis. As far as I know, different meta learning algorithms lead to (sometimes vastly) different final results. Traditionally, it's known that scaling iterative-differentiation (as opposed to implicit differentiation) meta learning algorithms to large models/datasets is bottlenecked by prohibitive compute/memory costs. Since this paper addresses this issue, it would be nice if meta-learning algorithm implemented with their method actually improves the final performance.

**Methods And Evaluation Criteria:**

The main focus of the paper is improving the implementation of meta-learning (or bi-level optimization) algorithms. I believe all proposed implementation tricks intuitively make sense and smartly designed.

**Other Comments Or Suggestions:**

I believe this paper sits between ML and systems paper. Given that ICML has been traditionally closer to ML conference, I am not sure how system contributions are received in this avenue.

**Other Strengths And Weaknesses:**

Using HVP or JVP to enable efficient bi-level optimization is a popular idea itself. However, the key contributions of this paper are practical implementation tricks for efficient HVP/JVP.

**Questions For Authors:**

N/A

**Relation To Broader Scientific Literature:**

Bi-level optimization with recent large models, despite having lots of useful applications like hyper-parameter optimization, is largely limited by its compute/memory complexity. This paper tackles this issue, potentially providing new research opportunities.

**Theoretical Claims:**

N/A: The paper doesn't make any theoretical claims besides providing derivations of their MixFlow-MG algorithm.

---

> ### Author Rebuttal · Authors · 2025-03-31
>
> We appreciate the feedback of the Reviewer and address the raised concerns below.
>
>
> > The only issue I see is a lack of the meta learning performance analysis. As far as I know, different meta learning algorithms lead to (sometimes vastly) different final results.
>
> We agree with the Reviewer that the landscape of meta learning algorithms is truly vast. In this paper, we focus on the class of gradient-based backpropagation through time (BPTT) optimisation methods and, in particular, on their computational and memory aspects. The proposed algorithm MixFlowMG helps to scale the existing practical setups up by more than an order of magnitude, whilst remaining theoretically identical (up to machine precision) to the well-studied standard BPTT methods, as we demonstrated in Section 3 and Appendix A.2.
>
> In particular, this implies that the relevant results from the works that use this class of algorithms are fully transferable to MixFlowMG. To double-check this, we validated that all our benchmarks produce the same numerical results as the baselines, with up to 1e-6 absolute and 1e-4 relative errors (which is within the machine precision for 32-bit floating point arithmetics).
>
> We would like to highlight that the proposed algorithm can be combined with a wide variety of meta-learning algorithms, with the same guarantee of equivalence (up to numerical precision), as well as with future algorithms that have not yet been discovered.
>
> > … it would be nice if meta-learning algorithm implemented with their method actually improves the final performance.
>
> We would like to highlight that several works, e.g. [1], [2], [3], reported positive effects of increasing the number of inner steps (up to 10 in [1], [2], and up to 100 in [3]) in the BPTT-based meta optimisation. These results, combined with the theoretical equivalence of MixFlowMG to the presented algorithms, keeps us optimistic that increasing this parameter by over 10 times with MixFlowMG is likely to follow this positive trend, potentially leading to qualitatively new outcomes. That being said, to ensure the well-defined scope of the paper, we leave such investigations to future works and instead devote the experimental section mainly to the thorough technical analysis of the proposed algorithm.
>
> [1] _Figure 5_ in Finn, Chelsea, Pieter Abbeel, and Sergey Levine. "Model-agnostic meta-learning for fast adaptation of deep networks." International conference on machine learning. PMLR, 2017
>
> [2] _Table 2_ in Antoniou, Antreas, Harrison Edwards, and Amos Storkey. "How to train your MAML." arXiv preprint arXiv:1810.09502 (2018)]
>
> [3] _Figure 2a_ in Rajeswaran, Aravind, et al. "Meta-learning with implicit gradients." Advances in neural information processing systems 32 (2019)
>
> > I believe this paper sits between ML and systems paper. Given that ICML has been traditionally closer to ML conference, I am not sure how system contributions are received in this avenue.
>
> We consider the main contribution of this paper to be a novel algorithm to reduce the computational requirements of meta-gradients based on mixed-mode differentiation.
>
> ICML conference lists the following category in the call-for papers https://icml.cc/Conferences/2025/CallForPapers:  Machine Learning Systems (improved implementation and scalability, hardware, libraries, distributed methods, etc.). We believe that this work falls under this category and that it can be of interest to the ML community, as it can be implemented and used by people right away. We will also include an implementation of MixFlowMG in PyTorch in the final version of the paper.
>
> Some examples of similar conference papers, including ones from the past ICML conferences:
>
> * Archer, Aaron, et al. "Practical Performance Guarantees for Pipelined DNN Inference." International Conference on Machine Learning. PMLR, 2024. https://proceedings.mlr.press/v235/archer24a.html
>
> * Biderman, Stella, et al. "Pythia: A suite for analyzing large language models across training and scaling." International Conference on Machine Learning. PMLR, 2023. https://proceedings.mlr.press/v202/biderman23a.html
>
> * Patil, Shishir G., et al. "POET: Training neural networks on tiny devices with integrated rematerialization and paging." International Conference on Machine Learning. PMLR, 2022. https://proceedings.mlr.press/v162/patil22b/patil22b.html
>
> * Dao, Tri, et al. "Flash Attention: Fast and memory-efficient exact attention with io-awareness." Advances in neural information processing systems 35 (2022): 16344-16359. https://openreview.net/pdf?id=H4DqfPSibmx
>
> * Stanton, Samuel, et al. "Kernel Interpolation for Scalable Online Gaussian Processes." International Conference on Artificial Intelligence and Statistics. PMLR, 2021. https://proceedings.mlr.press/v130/stanton21a.html

---

### Decision · Program_Chairs · 2025-05-01

**Decision:**

Accept (poster)

**Comment:**

The paper proposes a method (MixFlow-MG) to reduce the memory consumption when computing exact meta-gradients of bi-level optimization problems whose inner loop uses a gradient-based optimizer.
This is achieved by re-parameterizing the inner loop which allows to replace double-backward differentiation with mixed-mode differentiation (forward-over-reverse) that scales more favourably in terms of memory.
Empirical results demonstrate memory reductions by up to 10x as well as small run time improvements on three different setups:

- The proposed method is simple and compatible with the parameterizations assumed by many libraries, allowing it to be broadly adopted.
  The submission contained a JAX implementation, and the authors will add a PyTorch implementation, further broadening applicability.
  Most reviewers appreciated this contribution and emphasized that bi-level optimization research is often limited by memory.
  The proposed method's achieved memory reductions allow existing works to scale by a factor of 10; this is a solid contribution.

- There were concerns regarding the motivation of improving the **exact** computation of meta-gradients, and the lacking comparison with approximate schemes.
  I understand this concern, but think that successfully scaling the exact computation in itself is a sustainable contribution that might lead to the development of new approximate schemes that inherit the proposed improvements.
  The authors somewhat addressed this in their rebuttal and provided related work that suggests improved results by scaling the exact scheme, e.g. using longer inner loops.
  After reading the paper, I believe that the proposed approach can also benefit some existing approximate schemes, e.g. truncated backpropagation through time.
  Finally, the experiments presented in the paper focus on improving computational performance and do not show how MixFlow-MG improves performance down-stream.
  I think this would be good a addition for the paper, but secondary for its acceptance as this should follow from the demonstrated gains.

Overall, the paper is in good shape and the authors have offered concrete actions to further improve it; therefore I recommend acceptance.